# Relaxation of synaptic inhibitory events as a compensatory mechanism in fetal SOD spinal motor networks

Pascal Branchereau*, Elodie Martin, Anne-Emilie Allain, William Cazenave, Laura Supiot, Fara Hodeib, Amandine Laupénie, Urvashi Dalvi, Hongmei Zhu, Daniel Cattaert

University of Bordeaux, CNRS, INCIA, UMR 5287, Bordeaux, France

**Abstract** Amyotrophic lateral sclerosis (ALS) is a devastating neurodegenerative disease affecting motor neurons (MNs) during late adulthood. Here, with the aim of identifying early changes underpinning ALS neurodegeneration, we analyzed the GABAergic/glycinergic inputs to E17.5 fetal MNs from SOD1$^{G93A}$ (SOD) mice in parallel with chloride homeostasis. Our results show that IPSCs are less frequent in SOD animals in accordance with a reduction of synaptic VIAAT-positive terminals. SOD MNs exhibited an $E_{GABAAR}$ 10 mV more depolarized than in WT MNs associated with a KCC2 reduction. Interestingly, SOD GABAergic/glycinergic IPSCs and evoked GABA$_A$R-currents exhibited a slower decay correlated to elevated $[Cl^-]_i$. Computer simulations revealed that a slower relaxation of synaptic inhibitory events acts as compensatory mechanism to strengthen GABA/glycine inhibition when $E_{GABAAR}$ is more depolarized. How such mechanisms evolve during pathophysiological processes remain to be determined, but our data indicate that at least SOD1 familial ALS may be considered as a neurodevelopmental disease.

## Introduction

Amyotrophic Lateral Sclerosis (ALS), also known as Lou Gehrig's disease, is a rapidly progressive neurodegenerative disease that targets motor neurons (MNs). It is one of the most common and most devastating neurodegenerative diseases. The incidence rate of ALS in the general European population has been estimated as ~2 per 100 000 person per year (*Logroscino et al., 2010*). What makes ALS particularly devastating is that there is no known curative treatment. Death often occurs due ultimately to respiratory paralysis that concludes the progressive pattern of the disease, with the mean survival time of patients post-diagnosis being 3 to 5 years.

In 90% of cases, ALS is idiopathic and sporadic (sALS). Only 10% of ALS cases are familial (fALS) in origin, inherited through an autosomal dominant pattern. Around 20 genes are associated with ALS, with the most common causes of typical ALS being associated with mutations in SOD1, TARDBP, FUS and C9orf72 (*Van Damme et al., 2017*). Mutations in C9orfF72 are the most common cause of fALS (10% of total ALS) followed by mutations in SOD1 (superoxide dismutase 1) (2% of total ALS). Interestingly, misfolded SOD1 was found in MNs in a subset of patients with sALS that did not have *SOD1* mutations, suggesting that there is a SOD1-dependent pathway common to both sALS and fALS (*Bosco et al., 2010*; *Paré et al., 2018*).

Because ALS is a lethal adult-onset neurodegenerative disease, most studies have focused on symptomatic adult stages. However, a growing body of evidence indicates that ALS pathogenesis might develop earlier than previously expected (*Kuo et al., 2004*; *Kuo et al., 2005*; *Avossa et al., 2006*; *Bories et al., 2007*; *Amendola and Durand, 2008*; *van Zundert et al., 2008*; *Pambo-Pambo et al., 2009*; *Chang and Martin, 2011*; *Quinlan et al., 2011*; *Filipchuk and Durand, 2012*; *Leroy et al., 2014*; *Milan et al., 2015*; *Chang and Martin, 2016*; *Medelin et al., 2016*). Using the

*For correspondence:
pascal.branchereau@u-bordeaux.fr

Competing interests: The authors declare that no competing interests exist.

transgenic mouse model SOD1$^{G93A}$ (SOD, Gly93→Ala substitution), which expresses a large degree of human mutant SOD1 and faithfully recapitulates a vast majority of the pathology's abnormalities seen in ALS patients (*Fogarty, 2018*), we previously found that SOD MNs are hyperexcitable at the prenatal (embryonic day (E) 17.5) stage because of a shorter dendritic tree and increased input resistance (*Martin et al., 2013*).

Among the numerous potential causative link in ALS associated dysfunction and degeneration, more and more evidence point to the key role of inhibitory alteration (*Ramírez-Jarquín et al., 2014*; *Clark et al., 2015*; *Ramírez-Jarquín and Tapia, 2018*; *Van den Bos et al., 2018*), with vulnerability being induced in the perinatal period (*Eisen et al., 2014*).

Here, using gramicidin intracellular recordings of SOD and wild type (WT) lumbar spinal prenatal E17.5 MNs from the lateral motor column (LMC, ventral pool), we first analyzed chloride homeostasis and found that $E_{GABAAR}$ was 10 mV more depolarized in the SOD fetus compared to WT associated with a KCC2 down-regulation. Interestingly, locomotor-like activity remained normal. We then examined the premotor network at the same fetal developmental stage in order to determine underlying mechanisms that could compensate for MN hyperexcitability and chloride homeostasis deficiency. This was performed by combining whole-cell patch-clamp recordings and immunohistochemistry. Miniature (m), spontaneous (s) inhibitory postsynaptic currents (IPSCs) as well as evoked (e) IPSCs and puff-evoked GABA$_A$R currents were quantified in parallel with the staining of synaptic VIAAT boutons. Our results show a significant reduction in IPSC frequency in SOD MNs compared to WT MNs, in agreement with a reduction in synaptic VIAAT-positive terminals. Our result also show a significantly slower decay time in SOD IPSCs and GABA$_A$R-currents compared to WT. Interestingly, this slower decay is associated with a higher intracellular chloride concentration $[Cl^-]_i$ in SOD MNs compared to WT. In prenatal E17.5 MNs, GABAergic/glycinergic postsynaptic potentials are depolarizing (dGPSPs) and exert mixed excitatory (depolarizing) and inhibitory (shunting) effects on the input-output (I-O) relationship (*Gulledge and Stuart, 2003*; (*Alger and Nicoll, 1982*; *Michelson and Wong, 1991*; *Misgeld et al., 1982*; *Staley and Mody, 1992*). Higher $[Cl^-]_i$ would then result in shifting the balance dGPSPs toward more excitation (*Branchereau et al., 2016*). However, Using computer simulations, we demonstrate that longer decay time reinforce the inhibitory component of dGPSPs. Therefore longer relaxation of dGPSPs constitutes a partial compensatory mechanism sustaining a well-coordinated, although slightly slower, locomotor activity in prenatal SOD mice. Together, our findings show that although the presynaptic network of SOD MNs is altered at fetal stages, cellular mechanisms leading to long-lasting IPSCs operate to compensate for the depolarized $E_{GABAAR}$ and hyperexcitability of E17.5 SOD MNS in order to maintain a coordinated locomotor activity.

## Results

### Fetal MNs exhibit a perturbed chloride homeostasis but normal locomotor-like activity

Initially, gramicidin perforated patch-clamp recordings from L4-L5 MNs were used to obtain Rin and Cm values for individual MNs. SOD MN values for Rin and Cm were 148.2 ± 13.5 MΩ, n = 21 and 105.6 ± 5.9 pF, n = 21, N = 16, respectively, and were significantly different (p<0.05, Mann Whitney test) from WT MNs values (104.9 ± 10.8 MΩ, n = 16 and 143.7 ± 14.2 pF, n = 16, N = 10) (*Table 1*). These values corresponded to the previously reported differences between WT and SOD MNs (*Martin et al., 2013*).

**Table 1.** Passive membrane properties of E17.5 MNs and GABA$_A$R conductances.
Data are from perforated patch-clamp recordings and are shown as mean ± SEM. Statistical significance was calculated by a non-parametric Mann Whitney test. ns, p>0.05; *p<0.05.

| | Rin (MΩ) | Cm (pF) | g$_{GABAAR}$ (nS) | g$_{GABAAR}$/Cm (nS/pF) | N | N |
|---|---|---|---|---|---|---|
| WT | 104.9 ± 10.8 | 143.7 ± 14.2 | 3.90 ± 0.97 | 0.029 ± 0.007 | 16 | 10 |
| SOD | 148.2 ± 13.5* | 105.6 ± 5.9* | 1.91 ± 0.24[ns] | 0.019 ± 0.003[ns] | 21 | 16 |

Puff application of the GABA$_A$R agonist isoguvacine allowed us to measure the reversal potential of GABA$_A$R (E$_{GABAAR}$) in individual SOD and WT MNs (*Figure 1A*), while our pooled data indicated that mean E$_{GABAAR}$ was significantly different in SOD MNs (−50.5 ± 2.4 mV, n = 25, N = 18) and WT MNs (−62.0 ± 2.0 mV, n = 24, N = 17) (p<0.001, Mann Whitney test) (*Figure 1B1*). From these E$_{GABAAR}$ values, and because Cl$^-$ ions are mainly involved in GABA$_A$R currents in fetal MNs (*Bormann et al., 1987*; *Gao and Ziskind-Conhaim, 1995*), we calculated the intracellular chloride concentration [Cl$^-$]$_i$ as being 19.5 ± 1.5 mM and 12.2 ± 0.9 mM in SOD and WT MNs, respectively (p<0.001, Mann Whitney test) (*Figure 1B2*). Interestingly, [Cl$^-$]$_i$ values reported in E17.5 WT MNs are in agreement with those previously reported, i.e., 12 mM (*Delpy et al., 2008*). The resting membrane potential (E$_m$) was not significantly different between SOD and WT MNs (−73.1 ± 0.5 mV in SOD and −74.1 ± 1.0 mV in WT MNs (p>0.05, Mann Whitney test MNs) (*Figure 1B3*), in contrast to the mean driving force (DF = E$_m$ - E$_{GABAAR}$), which was 22.6 ± 2.3 mV and 12.1 ± 2.2 mV in SOD and WT MNs, respectively (p<0.01, Mann Whitney test) (*Figure 1B4*).

The dissipation of an elevated intracellular Cl$^-$ concentration is mediated by the activation of the neuronal K$^+$/Cl$^-$ co-transporter type 2 (KCC2) (*Payne et al., 1996*; *Rivera et al., 1999*). This protein, which extrudes Cl$^-$, is expressed at early stages in the embryonic SC (*Delpy et al., 2008*). The [Cl$^-$]$_i$ is also related to the Na$^+$-K$^+$−2Cl$^-$ co-transporter NKCC1, which intrudes chloride ions (*Russell, 2000*; *Delpy et al., 2008*). Therefore, in order to identify the cellular mechanisms underlying the higher [Cl$^-$]$_i$ in SOD MNs, we firstly assessed the amount of KCC2 and NKCC1 in lumbar SCs using WB. Our data showed that KCC2 was reduced by ~19% in SOD lumbar SCs compared to WT (p<0.05, Mann Whitney test) (*Figure 2A1*) whereas NKCC1 was unchanged (*Figure 2A2*). When analyzed in the MN area (Hb9-eGFP), the KCC2 staining appeared as significantly reduced in SOD SCs (181.5 ± 16.2 arbitrary unit (AU), n = 43, N = 4) compared to WT (395.0 ± 20.3 AU, n = 42, N = 4) (p<0.001, Mann Whitney test) (*Figure 2B1–B2 and B5*). At the level of the MN membrane, the KCC2 density was also significantly reduced in SOD SCs (5.8 ± 0.6 AU, n = 81, N = 4) compared to WT SCs (8.5 ± 0.6 AU, n = 116, N = 4) (p<0.01, Mann Whitney test) (*Figure 2B3–B4 and B6*). Functionally, we found that the KCC2 efficacy, assessed using indirect measurements, was reduced in SOD MNs: E$_{GABAAR}$ reached a maximum of −43.8 ± 0.8 mV (n = 19, N = 16) in SOD MNs and −46.5 ± 0.7 mV (n = 24, N = 17) in WT (p<0.05, Mann Whitney test) after 30 s isoguvacine pressure ejection (*Figure 2C*) and returned to control values: −49.3 ± 1.2 mV (n = 19) in SOD MNs and −54.8 ± 1.0 mV (n = 24) in WT (p<0.001, Mann Whitney test) after a 4 min isoguvacine washout (*Figure 2C*). This difference was absent when the specific KCC2 blocker VU0240551 was applied (*Figure 2C*).

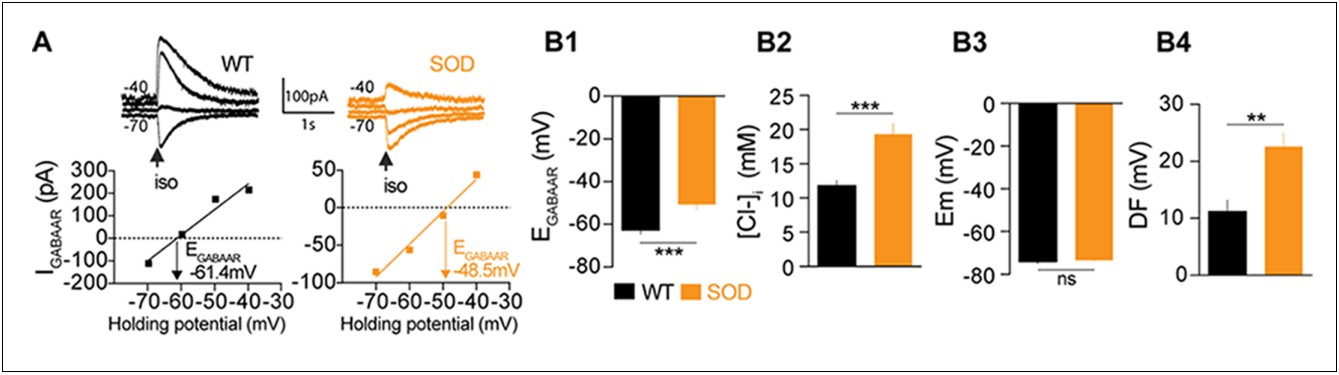

**Figure 1.** Alterations in chloride homeostasis. (**A**) Representative traces of isoguvacine (Iso) responses illustrating the reversal of the evoked GABA$_A$R-related current (E$_{GABAAR}$) in WT and SOD MNs. Holding voltage −70 mV, −60 mV, −50 mV and −40 mV. I/V relationships, plotted below the traces, revealed that E$_{GABAAR}$ was −61.4 mV and −48.5 mV in the representative WT and SOD MNs, respectively. (**B1**) Mean E$_{GABAAR}$ was significantly lower in E17.5 WT (n = 24) compared to SOD (n = 25) MNs from the same litter. (**B2**) Accordingly, the mean [Cl$^-$]$_i$ calculated from individual E$_{GABAAR}$ values, indicated that SOD MNs had a higher [Cl$^-$]$_i$ than WT. (**B3**) Mean resting membrane potential (Em) did not differ in WT and SOD MNs. (**B4**) Mean driving force (DF) for chloride ions was doubled in SOD MNs compared to WT MNs. Values illustrated were from the same set of MNs. ***p<0.001,**p<0.01, *ns* non-significant, Mann Whitney test.

The online version of this article includes the following figure supplement(s) for figure 1:

**Figure supplement 1.** Recorded MNs belong to the lateral motor column (LMC).

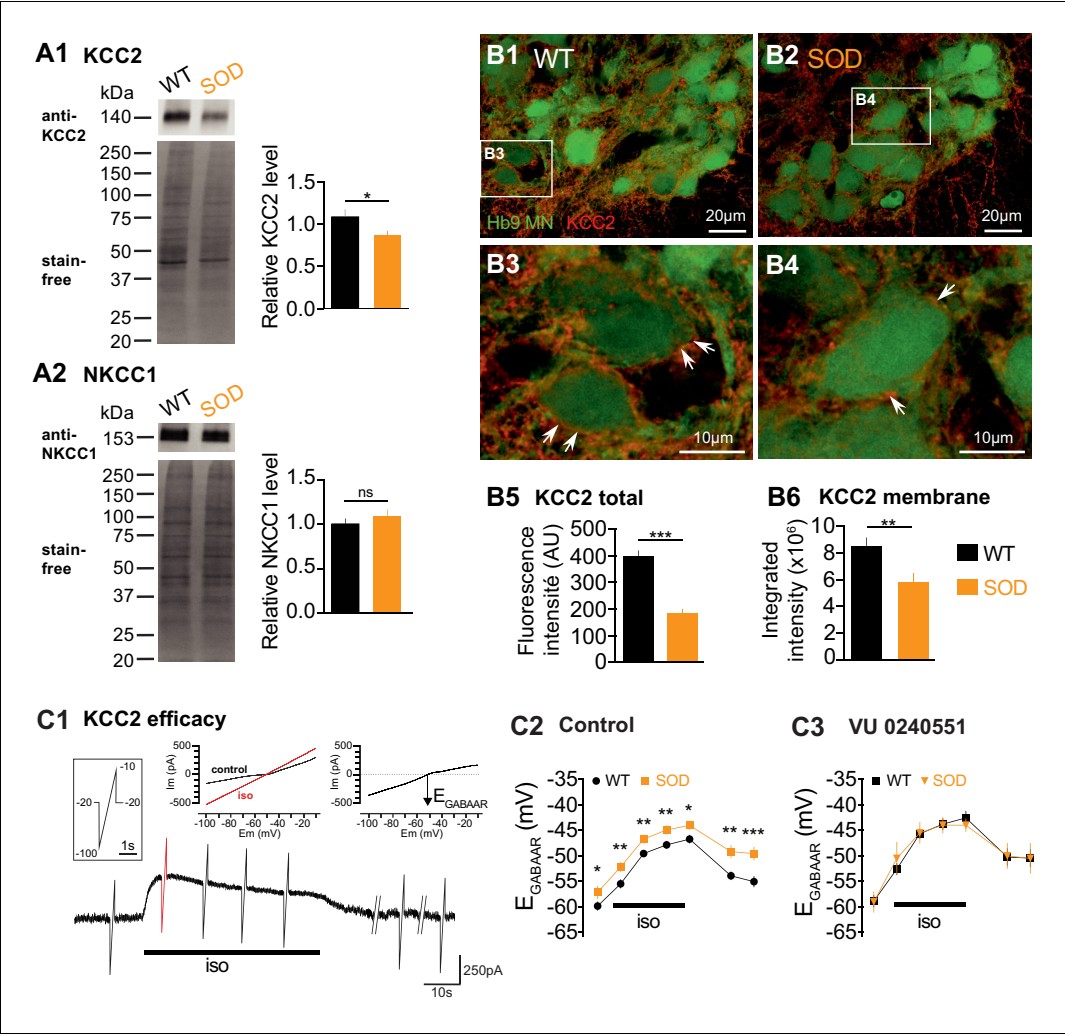

**Figure 2.** Expression of KCC2 and NKCC1 in the E17.5 SOD spinal cord. (**A1**) Analysis of the KCC2 protein in SOD and WT SCs. The stain-free staining of total proteins loaded (lower left panel) was used as the normalization control to quantify the ~140 kDa KCC2 band (upper left panel). Right panel: quantification of KCC2 WB. (**A2**) Analysis of the NKCC1 protein in SOD and WT SCs. The ~150 kDa NKCC1 band (upper left panel) was analyzed. Quantification of NKCC1 WB. Three loads (20 μg of protein) from SOD and WT lumbar spinal cord extracts were used, each load including five individual SCs from four different litters. (**B1–B4**) Representative immunohistochemical KCC2 (red) staining in WT (**B1–B3**) and SOD (**B2–B4**) lumbar 0.2 μm thickness optical sections (L3-L5 level) containing Hb9-eGFP MNs (green). Images correspond to x60 confocal acquisitions. White arrows point the KCC2 labeling surrounding spinal MNs. (**B5–B6**) Quantification of the global KCC2 staining (**B5**) in the MN area and membrane KCC2 staining (**B6**) in E17.5 SOD and WT SCs. (**C1**) Illustration of the protocol applied to assess the KCC2 efficacy as explained in the Materials and method section. (**C2–C3**) Evolution of $E_{Cl}$ values with time during the isoguvacine (iso) application in control aCSF (**C2**) and in the presence of the highly specific KCC2 blocker VU0240551 (10 μM) (**C3**). ***$p<0.001$, **$p<0.01$, *$p<0.05$, *ns* non-significant, Mann Whitney test.

In order to assess whether the alteration of chloride homeostasis in SOD MNs could affect the left-right alternation in locomotor-like activity expressed by E17.5 mouse spinal lumbar networks due to half-centers organization and commissural inhibition (*Branchereau et al., 2000*), we recorded this activity in L3-L5 ventral roots after exogenous application of 5-HT/NMDA/DA. As illustrated in *Figure 3A1*, this cocktail evoked stable bilateral segmental alternation between left and right ventral roots in WT SC preparations. Chemically-activated locomotor-like activity was also expressed in the ventral roots of SOD SCs (*Figure 3A2*), although analysis (Rayleigh's test) revealed a slower rhythm period in SOD SCs (2.01 ± 0.03 s, n = 8, N = 5) compared to WT controls where bursts recurred with

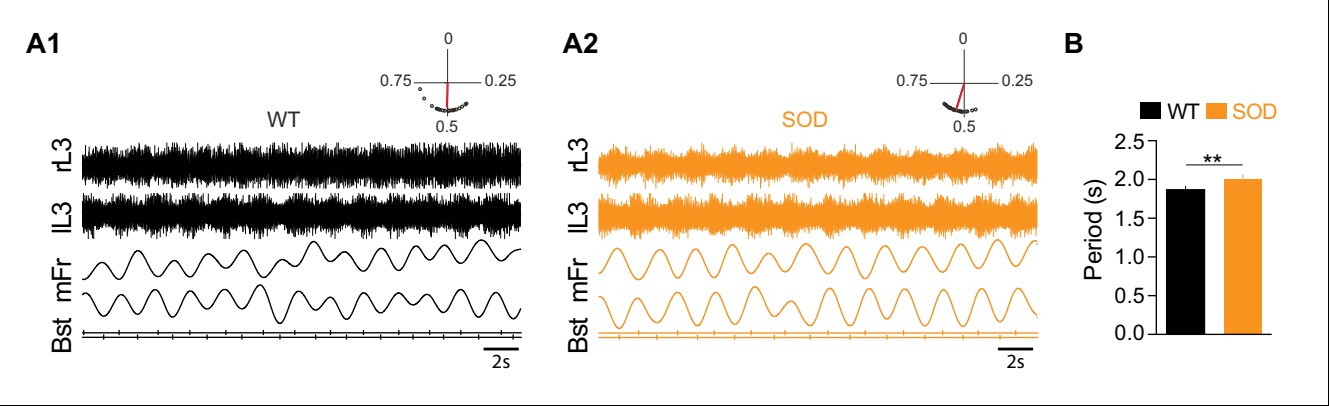

**Figure 3.** Motor activity in the E17.5 SOD spinal cord. (**A**) Fictive locomotor-like activity evoked by 5-HT/NMDA/DA (see Materials and methods) recorded from contralateral lumbar ventral roots (lL3, rL3; two first traces) displayed regular alternating activities (see circular plots above each raw data panel) in both WT (**A1**) and SOD (**A2**) SCs. Below the raw data are traces of the floating mean spike frequencies (mFr) and the corresponding troughs of activity (burst start, Bst) from which the phase relationships between the left and right SC sides were calculated (circular plots). (**B**) The cycle period was significantly longer in SOD SCs (n = 9) compared to WT (n = 8) SCs. **p<0.01, *ns* non-significant, Mann Whitney test.

a period of 1.88 ± 0.03 s (n = 9, N = 7) (p<0.01, Mann Whitney test) (*Figure 3B*). However, the rhythm phase relationship between left and right sides was close to anti-phase in both genotypes (0.49 ± 0.003 and 0.52 ± 0.004 in SOD and WT SCs, respectively) (see polar plots in *Figure 3A1–A2*). Therefore, surprisingly, alteration of chloride homeostasis in SOD MNs has limited consequences on the locomotor-like activity and therefore little physiological consequences. Could compensatory mechanisms in the SOD spinal synaptic network explain this result?

In order to gain information about the synaptic inputs from the premotor inhibitory neuronal networks that underlies this MN activity, we then compared the occurrence and properties of GABA-/glycinergic synaptic events in SOD and WT E17.5 MNs in the absence of rhythmic fictive locomotion. We found that miniature GABAergic/glycinergic inhibitory synaptic currents (mIPSCs), which persisted in the presence of TTX, were significantly (p<0.001; K–S test) less frequent in SOD: the mean inter-event interval (IEI) was 887 ± 30 ms (n = 19, N = 9, 1589 events) for SOD MNs and 838 ± 33 ms (n = 15, N = 7, 1366 events) for WT MNs from the same littermate (p<0.001, Mann Whitney test) (*Figure 4A1,A2*). The mIPSCs of SOD MNs were also slightly smaller (52.8 ± 0.9 pA) compared to WT MNs (57.9 ± 1.2 pA) (p<0.05, Mann Whitney test) (*Figure 4A3*). $Tau_{rise}$ was not significantly different between SOD MNs (1.25 ± 0.03 ms) and WT MNs (1.11 ± 0.02 ms) (p>0.05, Mann Whitney test), whereas $tau_{decay}$ was strongly significantly larger in SOD MNs (20.53 ± 0.08 ms) compared to WT MNs (16.11 ± 0.40 ms) (p<0.0001, Mann Whitney test) (*Figure 4A4*, see inset traces). The total charge of mIPSCs, calculated for best-fitted (R > 0.98) mIPSCs, was similar in SOD MNs (−1.14 ± 0.04 pA.s, n = 841, N = 9) and WT MNs (−1.18 ± 0.04 pA.s, n = 901, N = 7) (p>0.05, Mann Whitney test), indicating that the longer relaxation for SOD mIPSCs likely compensated their reduced amplitude. The development of synaptic inhibition in spinal MNs involves a functional switch between GABA to glycine events (*Gao et al., 2001*). We therefore focused on pharmacologically dissected GABA and glycine mIPSCs, isolated using strychnine and GABAzine, respectively. Interestingly, strychnine application revealed a tonic glycine current in 5/7 SOD MNs and 4/6 WT MNs (*Figure 4A5*) without any significant difference in amplitude between SOD MNs (11.72 ± 3.34 pA, n = 5, N = 5) and WT MNs (11.15 ± 2.36 pA, n = 4, N = 4) (p>0.05, Mann Whitney test). We found a tonic GABA current in only 3/8 SOD MNs and 2/7 WT MNs, this tonic current being comparable in both genotypes: 15.03 ± 1.09 pA (n = 3, N = 3) for SOD MNs and 10.40 ± 0.60 pA (n = 2, N = 2) for WT MNs (p>0.05, Mann Whitney test). This tonic glycine and GABA current likely favors tonic shunting effect as previously demonstrated (*Branchereau et al., 2016*; *Song et al., 2011*).

The pharmacological dissection revealed that mIPSCs were ~50% pure glycine,~40% pure GABA, the remaining events ~ 10% being mixed (see *Figure 4A6–A8*). No difference was found between SOD MNs (48.43 ± 8.26% glycine, n = 5, N = 5; 39.65 ± 16.77% GABA, n = 3, N = 3) and WT MNs (51.46 ± 9.41% glycine, n = 3, N = 3; 41.79 ± 7.97% GABA, n = 5, N = 5) (p>0.05, Mann Whitney

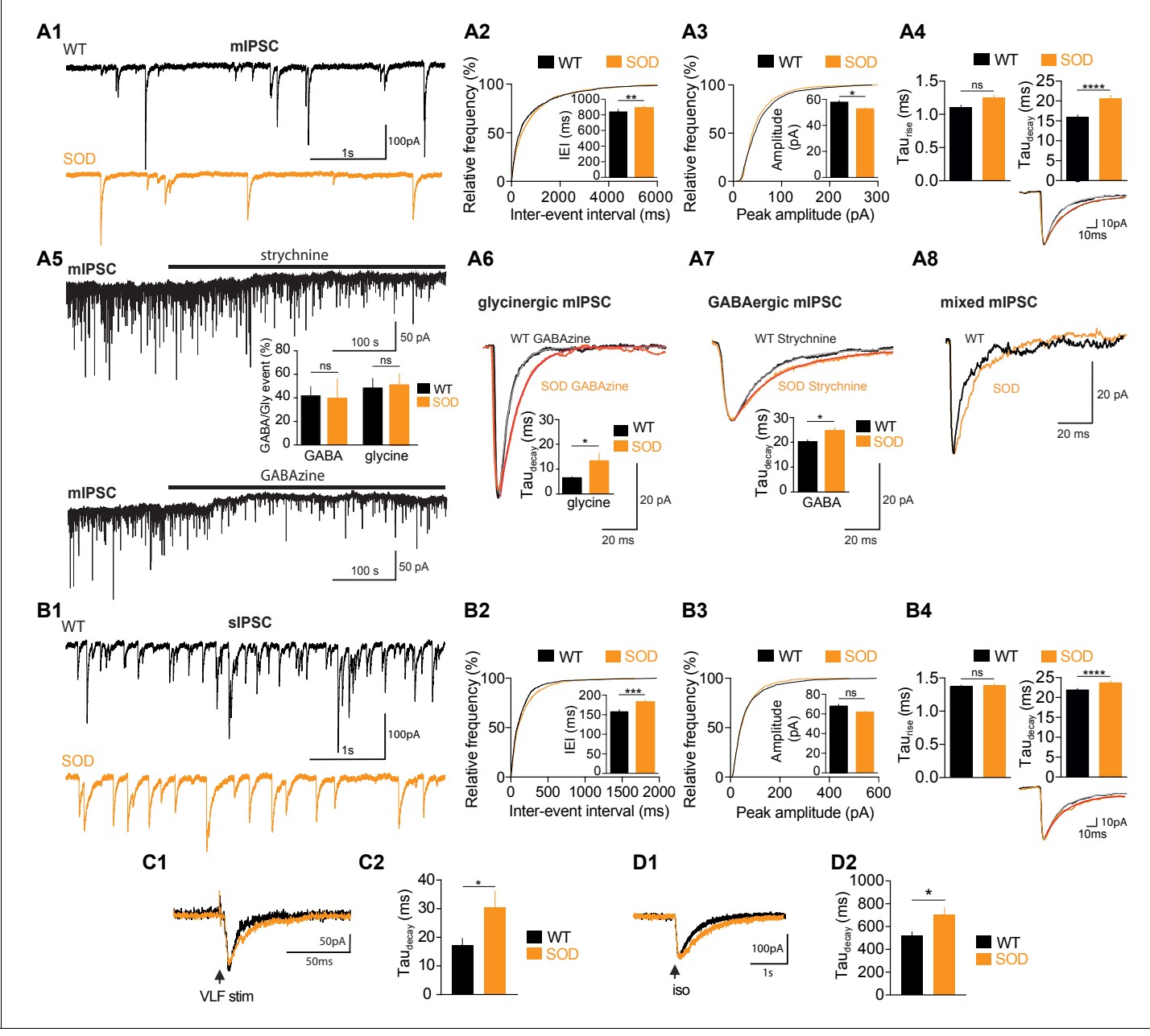

**Figure 4.** SOD E17.5 MNs exhibit a reduced IPSC frequency compared with WT MNs along with a persistent increase in their *tau_decay*. (**A1**) Sample miniature IPSC (mIPSC) recordings obtained from representative SOD and WT MNs (holding membrane potential −70 mV). (**A2**) Cumulative frequency of inter-event interval (IEI) and mean values of mIPSCS revealed a small but significant reduction in frequency in SOD MNs (n = 19) compared with WT (n = 15) MNs. (**A3**) mIPSC peak amplitude was slightly smaller in SOD than in WT MNs (same layout as in A2). (**A4**) Mean *tau_rise* was unchanged but mean *tau_decay* was significantly increased in SOD mIPSCs compared with WT mIPSCs. Inset traces (mean of 30 events) from representative experiments. (**A5**) Blocking the GlyR or GABA_AR (black horizontal bars) reveals a tonic current. The distribution of pure GABA_AR-mediated mIPSCs in WT (n = 5) and SOD (n = 3) MNs and pure GlyR-mediated mIPSCs in WT (n = 3) and SOD (n = 5) MNs is shown between the two traces. (**A6**) *Tau_decay* of pure GlyR-mediated mIPSCs is significantly increased in SOD MNs (n = 3) compared to WT MNs (n = 5). Traces are mean of 53 events (WT) and 57 events (SOD) from representative experiments. (**A7**) Relaxation of pure GABA_AR-mediated mIPSCs is significantly longer in SOD MNs (n = 3) compared to WT MNs (n = 5). Traces are mean of 35 events (WT) and 35 events (SOD) from representative experiments. (**A8**) Mixed GlyR-GABA_AR-mediated events from representative experiments. (**B1**) Samples of spontaneous IPSC (sIPSC) recordings from representative SOD and WT MNs. (**B2**) Cumulative frequency of IEI and mean values of sIPSCS revealed a slight frequency reduction in SOD MNs (n = 17) compared to WT MNs (n = 14). (**B3**) sIPSC peak amplitudes were similar in SOD and WT MNs. (**B4**) Mean *tau_rise* was not significantly different in WT sIPSCs, but SOD sIPSCs exhibited a higher *tau_decay*. Inset traces (mean of 30 events) from representative experiments. (**C1–C2**) SOD eIPSCs (n = 6) displayed a higher *tau_decay* than WT eIPSCs (n = 10). (**D1–D2**) Puff-evoked SOD GABA_AR currents (n = 21) exhibited a higher *tau_decay* than WT GABA_AR currents (n = 16). Left traces in **C1**) and **D1**) are from

*Figure 4 continued on next page*

*Figure 4 continued*

representative SOD and WT eIPSCs and puff-evoked GABA$_A$R currents. ****p<0.0001, ***p<0.001, **p<0.01, *p<0.05, *ns* non-significant, Mann Whitney test.

test). Interestingly, E17.5 SOD MNs, recorded at a holding potential of −70 mV with $E_{Cl}$ set at ~0 mV displayed pure glycine and GABA mIPSCs with a longer relaxation than E17.5 WT animals from the same littermate. $Tau_{decay}$ of pure glycine mIPSCs was 13.35 ± 3.14 ms (n = 3, N = 3) in SOD MNs and 6.53 ± 0.44 ms in WT MNs (n = 5, N = 5) (p<0.05, Mann Whitney test) (*Figure 4A6*) whereas $tau_{rise}$ were similar in both genotypes (1.24 ± 0.12 ms for SOD and 1.18 ± 0.07 ms for WT, p>0.05, Mann Whitney test). $Tau_{decay}$ of pure GABA mIPSCs was also larger in SOD MNs (24.69 ± 1.18 ms, n = 5, N = 5) compared to WT MNs (19.51 ± 0.56 ms, n = 3, N = 3) (p<0.05, Mann Whitney test) (*Figure 4A7*) whereas $tau_{rise}$ were similar in both genotypes (3.02 ± 0.24 ms for SOD and 2.12 ± 0.52 ms for WT, p>0.05, Mann Whitney test).

Most of the differences found between SOD and WT mIPSCs were also observed in spontaneous GABAergic/glycinergic inhibitory synaptic currents (sIPSCs). An increase in IEI was found for sIPSCs (p<0.001; K–S test): mean IEI was 182.3 ± 5.9 ms (n = 14, N = 10, 1499 events) and 157.9 ± 5.6 ms (n = 17, N = 11, 1685 events) in SOD and WT MNs, respectively (p<0.001, Mann Whitney test) (*Figure 4B1–B2*). No difference was found in the amplitude of sIPSCs (p>0.05; K-S test): 68.0 ± 1.8 pA in SOD MNs and 61.7 ± 1.4 pA in WT MNs (p>0.05, Mann Whitney test) (*Figure 4B3*). Again the $tau_{rise}$ was not significantly different between SOD MNs (1.38 ± 0.02 ms) and WT MNs (1.37 ± 0.02 ms) (p>0.05, Mann Whitney test), whereas $tau_{decay}$ was again strikingly larger in SOD MNs compared to WT MNs: 21.89 ± 0.42 ms in WT MNs and 23.58 ± 0.53 ms in SOD MNs (p<0.0001, Mann Whitney test) (*Figure 4B4*, see inset traces). VLF (ventro-lateral funiculus)-evoked IPSCs (eIPSCs) recorded in MNs at a holding potential −70 mV with an imposed $E_{Cl}$ of −45 mV also revealed a longer $tau_{decay}$ (30.3 ± 5.7 ms, n = 6, N = 3) in SOD MNs compared to WT MNs (18.3 ± 2.7, n = 10, N = 4) (p<0.05, Mann Whitney test) (*Figure 4C1–C2*), the amplitude and $tau_{rise}$ of eIPSCs being not significantly different (62.4 ± 16.6 pA, 2.8 ± 0.5 ms in SOD, n = 6 and 72.1 ± 13.2 pA, 2.3 ± 0.3 ms in WT, n = 10). This was in agreement with our data showing a reduced efficacy of KCC2 (*Figure 2C*) leading to a higher [Cl⁻]$_i$ in SOD MNs compared to WT and therefore a slower relaxation of IPSCs (*Pitt et al., 2008*; *Houston et al., 2009*). Finally, $tau_{decay}$ of puff-evoked GABA$_A$R current was also affected in the same way: 707.8 ± 57.8 ms (n = 21, N = 16) in SOD MNs *versus* 521.3 ± 33.3 ms (n = 16, N = 10) in WT (p<0.05, Mann Whitney test) (*Figure 4D1–D2*), the amplitude and $tau_{rise}$ of evoked IPSCs (eIPSCs) being not significantly different (49.3 ± 5.9 pA, 34.9 ± 3.6 ms in SOD, n = 21 and 58.1 ± 8.6 pA, 43.3 ± 5.8 ms, n = 16).

The transport of GABA/glycine (GABA/Gly) into synaptic vesicles is mediated by VIAAT transporters. Therefore, in order to seek for anatomical correlates of our electrophysiological findings, we performed an immunohistochemical analysis of GABAergic/glycinergic (VIAAT positive) synaptic puncta (*Figure 5A*, red) in the lumbar MN area identified by the presence of Hb9-eGFP neurons (*Figure 5A*, green). Synaptic terminals were identified using double staining for the synaptic vesicle protein synaptophysin (*Figure 5A*, white) that is ubiquitously expressed in the marginal zone of the SC in close proximity of Hb9 somata. We did not detect any significant difference in the percentage of VIAAT puncta between SOD and WT SCs: 0.47 ± 0.10% of analyzed area in SOD (n = 5, N = 4) and 0.54 ± 0.14% in WT (n = 6, N = 5) (p>0.05, Mann Whitney test) (*Figure 5B1–B2*). Synaptophysin staining in SOD SCs was decreased by 44% compared to WT: 2.17 ± 0.47% of area in SOD and 3.93 ± 0.72% of area in WT (p<0.01, Mann Whitney test) (*Figure 5B3*). We then calculated the percentage of the synaptophysin surface colocalized with VIAAT and found a ~ 23% decrease in this percentage of colocalization in SOD SCs compared to WT: 24.50 ± 2.55% and 32.05 ± 2.73% for SOD and WT, respectively (p<0.001, Mann Whitney test) (*Figure 5B4*).

Our electrophysiological data indicated a clear alteration in the frequency and shape of miniature and spontaneous synaptic currents in SOD MNs, with IPSCs being less frequent in SOD animals. This reduced basal network activity was therefore consistent with the anatomical labeling of GABAergic/glycinergic synaptic terminals. Prenatal SOD MNs also exhibited a more depolarized E$_{GABAAR}$ than WT MNs, which would in turn impinge on the efficacy of GABA/Gly inhibition (*Branchereau et al., 2016*). In SOD MNs E$_{GABAAR}$ was 10 mV more depolarized, which could preclude efficient inhibition. Surprisingly on the other hand, well-coordinated locomotor-like activity was still elicited in SOD SCs

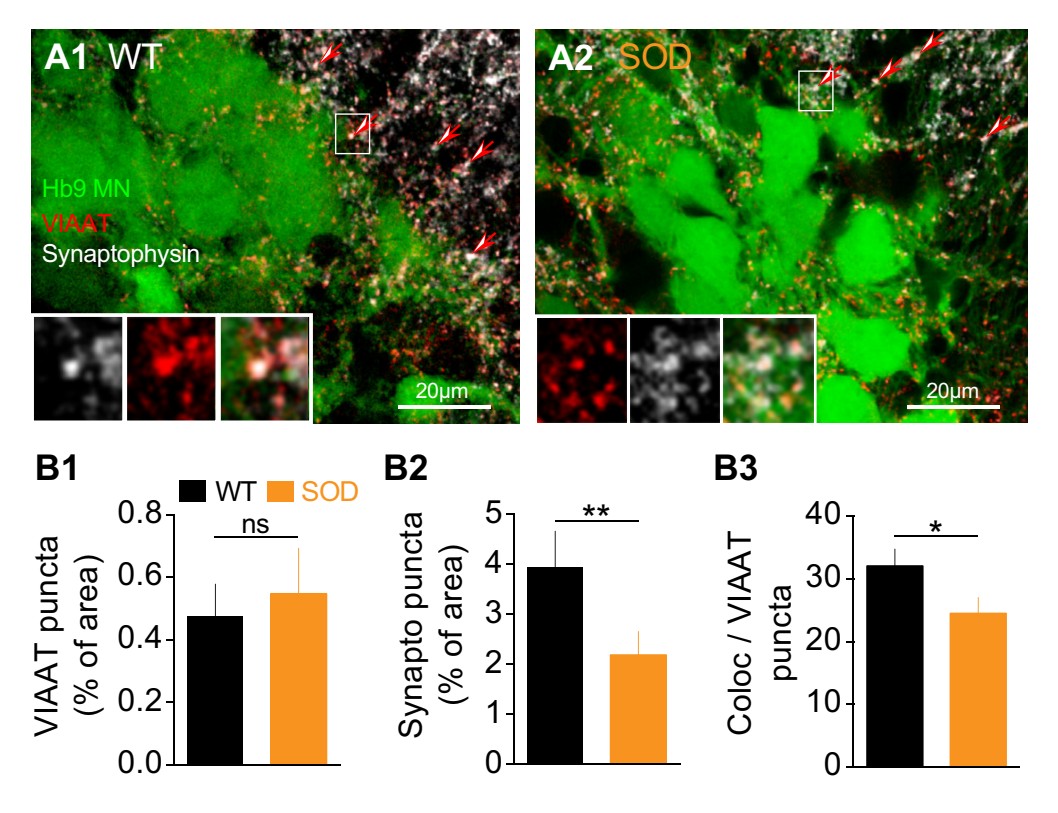

**Figure 5.** Inhibitory terminal staining in E17.5 SOD and WT SCs. (**A1-A2**) Confocal visualization of Hb9-eGFP MNs (green) along with VIAAT (red) and synaptophysin (white) staining in lumbar E17.5 WT (**A1**) and SOD1 SCs (**A2**) (frontal sections). Arrows point the synaptophysin-VIAAT co-localized staining highlighted in the left corner insets (0.2 µm thickness optical section). (**B1**) The density of VIAAT puncta was similar in WT (n = 6) and SOD1 (n = 5) SCs. (**B2**) SOD SCs again exhibited a significant reduction in their density of synaptophysin terminals. (**B3**) VIAAT puncta co-localizing with synaptophysin puncta were slightly reduced in SOD SCs compared with WT SCs. ***p<0.001, **p<0.01, *p<0.05, *ns* non-significant, Mann Whitney test.

(*Figure 3A*), indicating the presence of putative compensatory mechanisms. In this respect, the slower decay times of both IPSCs and puff-evoked GABA$_A$R currents recorded in SOD MNs could be one of such mechanisms. Therefore, in order to test this idea further, we conducted a series of computer simulations of SOD-like and WT-like E17.5 spinal MNs in which the effect of $tau_{decay}$ on the strength of GABA/Gly inhibition was assessed.

## A slower decay time of GABA/Gly synaptic currents to strengthen inhibition in SOD MNs

The impact of synaptic current shape on MN activity was assessed using numerical simulations of WT-like and SOD-like MNs with a NEURON simulation environment (*Carnevale and Hines, 2006*). The topology and morphometry of canonical WT and SOD E17 MNs were issued from *Martin et al. (2013)*. The two MN types differed only in the length of their terminal dendrites, being shorter in SOD-like MNs (i.e. 60% of WT-like MNs) (*Figure 6—figure supplement 1A-B*). As a result, the SOD-like MNs are more excitable (*Martin et al., 2013*). A continuous depolarizing current was injected into the somata of both the WT-like (250 pA) and SOD-like MNs (200 pA) (*Figure 6—figure supplement 1C*) to produce a spiking discharge of ~12.5 Hz (*Figure 6A1, B1*). This spike frequency in E17.5 MNs was reached during bursts of activity occurring during fictive locomotion (*Figure 6—figure supplement 2A, B*). During this MN discharge, a train of GABA/Gly synaptic events was delivered to the MN soma. Various frequencies of GABA/Gly synaptic events were tested to assess the frequency (cut-off frequency) needed to totally block the ongoing MN discharge. When E$_{GABAAR}$ was set to −50 mV (the value measured in biological SOD MNs), increasing the GABA/Gly $tau_{decay}$ from

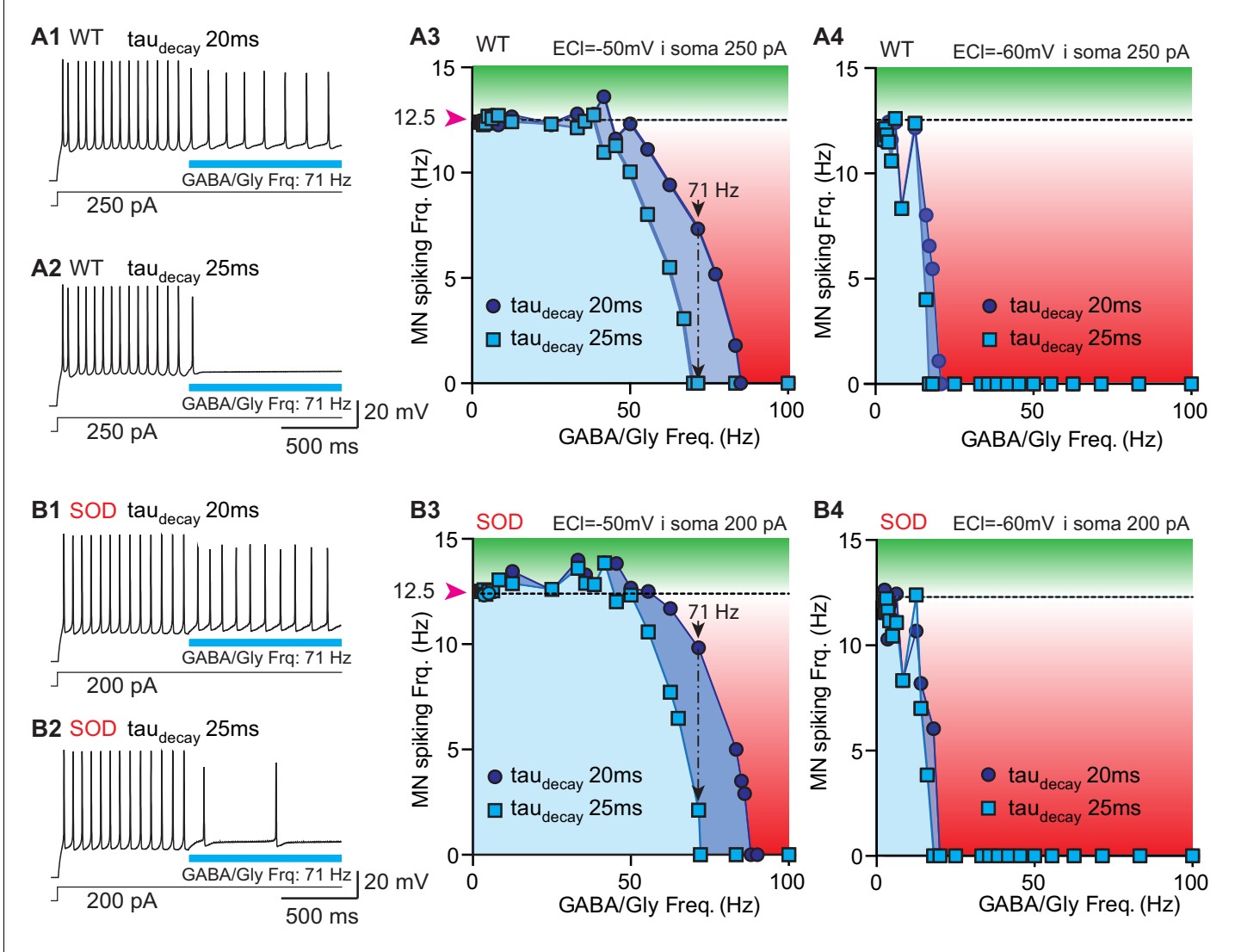

**Figure 6.** Simulation of a putative compensatory role of $tau_{decay}$ for the inhibitory strength of GABA/gly synaptic events in SOD-like MNs. (**A**) Simulations made with a WT-like MN whose soma was continuously injected with depolarizing current (250 pA) to induce spiking discharge at a rate of ~12.5 Hz. After the spiking discharge was stabilized (t = 2 s), a train of GABA/Gly synaptic events was delivered to the soma at a frequency of 71 Hz, with either a $tau_{decay}$ of 20 ms (**A1**) or 25 ms (**A2**) and $E_{GABAAR}$ set at −50 mV. Using these two values of GABA/Gly $tau_{decay}$ (20 ms: dark blue circles; 25 ms: light blue squares), various frequencies of GABA/Gly synaptic train (from 0 to 100 Hz) were tested with either $E_{GABAAR}$ = -50 mV (**A3**) or $E_{GABAAR}$ = −60 mV (**A4**). The horizontal dashed line represents the stabilized spiking frequency before application of the GABA/Gly synaptic train. Inhibitory effects (below dashed line) are in red, excitatory effects (above this line) are in green. (**B1–B5**) Simulations made with a SOD-like MN, in which the soma was continuously depolarized (200 pA injection) to induce spiking discharge at ~12.5 Hz (same layout as in **A**). Note that in the SOD-like MN with $E_{GABAAR}$ = −50 mV, excitatory effects are observed when the GABA/Gly synaptic frequency is below 50 Hz (**B3**). This effect is not observed when $E_{GABAAR}$ = −60 mV (**B4**) or is almost absent in the WT-like MN with $E_{GABAAR}$ = −50 mV.

The online version of this article includes the following figure supplement(s) for figure 6:

**Figure supplement 1.** Simulated E17.5 WT-like and E17.5 SOD-like MNs.

**Figure supplement 2.** Firing activity and synaptic inputs recorded from E17.5 MNs during fictive locomotion.

20 ms to 25 ms drastically increased its inhibitory effect on the MN discharge (*Figure 6*): whereas a 71 Hz GABA/Gly synaptic event train with a $tau_{decay}$ of 20 ms slowed down the WT model MNs' discharge from 12 Hz to 8–10 Hz, the same train totally blocked spike activity with a GABA/Gly $tau_{decay}$

of 25 ms (*Figure 6A1–A2*) and almost blocked it in a SOD MN (*Figure 6B1–B2*). We then repeated these simulations for a set of synaptic frequencies ranging from 0 Hz to 100 Hz for the two $tau_{decay}$ values (20 and 25 ms), with an $E_{GABAAR}$ set at −50 or −60 mV for the WT (*Figure 6A3–A4*) and SOD MNs (*Figure 6B3–B4*). The value of $E_{GABAAR}$ was determinant for the inhibitory effect of GABA/Gly events because in the WT-like MN, as in the SOD-like MN, the efficacy of the inhibitory train was much stronger when $E_{GABAAR}$ was set to −60 mV rather than −50 mV. Indeed, with $E_{GABAAR}$ = −60 mV, a 15 Hz GABA/Gly event train totally blocked the discharge in both MN types (*Figure 6A4–B4*), with little or no difference occurring between the two ($tau_{decay}$ of 20 and 25 ms) event/firing rate curves. By contrast, the corresponding curves differed significantly when $E_{GABAAR}$ was set to −50 mV (*Figure 6*, compare A3 with B3).

Significantly, the morphology of the MNs also played a role in the inhibitory effect of the 71 Hz GABA/Gly train on their activity with a $tau_{decay}$ set to 20 ms: the inhibition was stronger in the WT-like MN (discharge reduction from 12.5 Hz to 7.33 Hz, *Figure 6A3*) than in the SOD-like MN (reduction from 12.5 Hz to only 9.83 Hz, *Figure 6B3*). The effect of morphology was also evident in the difference (frequency shift) between the two curves ($tau_{decay}$ of 20 and 25 ms) being more pronounced in the SOD- (17 Hz) than in the WT-like MN (14 Hz). A final influence of MN morphology was observed for a GABA/Gly synaptic frequency below 50 Hz: in the SOD-like MN with $E_{GABAAR}$ = -50 mV, the GABA/Gly synaptic trains produced excitatory effects (*Figure 6B3*), although these effects were much smaller (if at all) in the WT-like MN (*Figure 6A3*). This excitatory action of GABA/Gly synapses was not observed when $E_{GABAAR}$ = −60 mV (*Figure 6A4–B4*).

We chose a firing frequency of 12.5 Hz for the model MNs since this corresponded to the mean value measured during locomotor-like activity in E17 WT embryos (*Figure 6—figure supplement 2A-B*). Together our results from computer simulations thus indicate that an increase of $tau_{decay}$ from 20 to 25 ms may constitute a partial compensatory mechanism contributing to reinforcing the inhibitory strength of GABA/Gly synaptic events that are dramatically reduced in SOD-type MNs mainly because of their more depolarized $E_{GABAAR}$ (−50 mV instead of −60 mV).

## Effect of slower decay time of GABA/Gly synaptic currents on locomotor rhythm

In a final step, we wished to assess the functional consequence of increasing $tau_{decay}$ during ongoing locomotor activity. To this end, a simplified computer simulation of two half-centers driving antagonistic MNs was employed (*Figure 7A*). Pacemaker activity was obtained using a simplified model (see Appendix 1) based on previous physiological observations on the fundamental role played by $I_{NaP}$ and $I_{KCa}$ in pacemaker activity (*Tazerart et al., 2008*). Interestingly, increasing $tau_{decay}$ slowed-down the locomotor rhythm when $E_{GABAAR}$ was set to −60 mV (WT MN value) and $g_{GABA/Gly}$ was set to 0.04 μS (*Figure 7A1, A2*). This slow-down of the locomotor rhythm was also observed when $g_{GABA/Gly}$ was increased to 0.2 μS with $E_{GABAAR}$ = −60 mV (*Figure 7B1*). When $E_{GABAAR}$ was set to −50 mV with a $g_{GABA/Gly}$ of 0.2 μS, increasing $tau_{decay}$ still slowed down the locomotor rhythm, but the effect was much smaller (*Figure 7B2*). This result was in agreement with our findings that the pharmacologically-evoked locomotor rhythm in SOD SCs was slower than in WT SCs (*Figure 3B*). Again, although this slight change in the locomotor rhythm observed in SOD SCs has likely no physiological consequences, it reveals one side effect of increasing $tau_{decay}$ of GABA/Gly synaptic events.

## Discussion

In the present study we show that fetal E17.5 SOD1[G93A] MNs express an impairment of chloride homeostasis that leads to a more depolarized reversal potential for GABA$_A$R ($E_{GABAAR}$), indicating that a very early inhibitory dysfunction may initiate the pathogenesis in ALS MNs as hypothesized by others (*van Zundert et al., 2012*; *Clark et al., 2015*). This could lead to a less efficient inhibitory input to MNs (*Branchereau et al., 2016*), which in turn would be expected to affect locomotor coordination. However, this was not observed here, likely due to a compensatory prolongation of inhibitory synaptic events revealed in our study. Moreover, we found that the passive properties of E17.5 MNs (Rin and Cm) complied with previous observations (*Martin et al., 2013*), in that Rin increased and Cm decreased in SOD MNs (*Table 1*), in accordance with a morphometric alteration (shorter dendritic length without changes at the soma level). Decreased capacitance and membrane conductance were therefore not related to smaller size of E17.5 SOD mouse MNs. These findings therefore

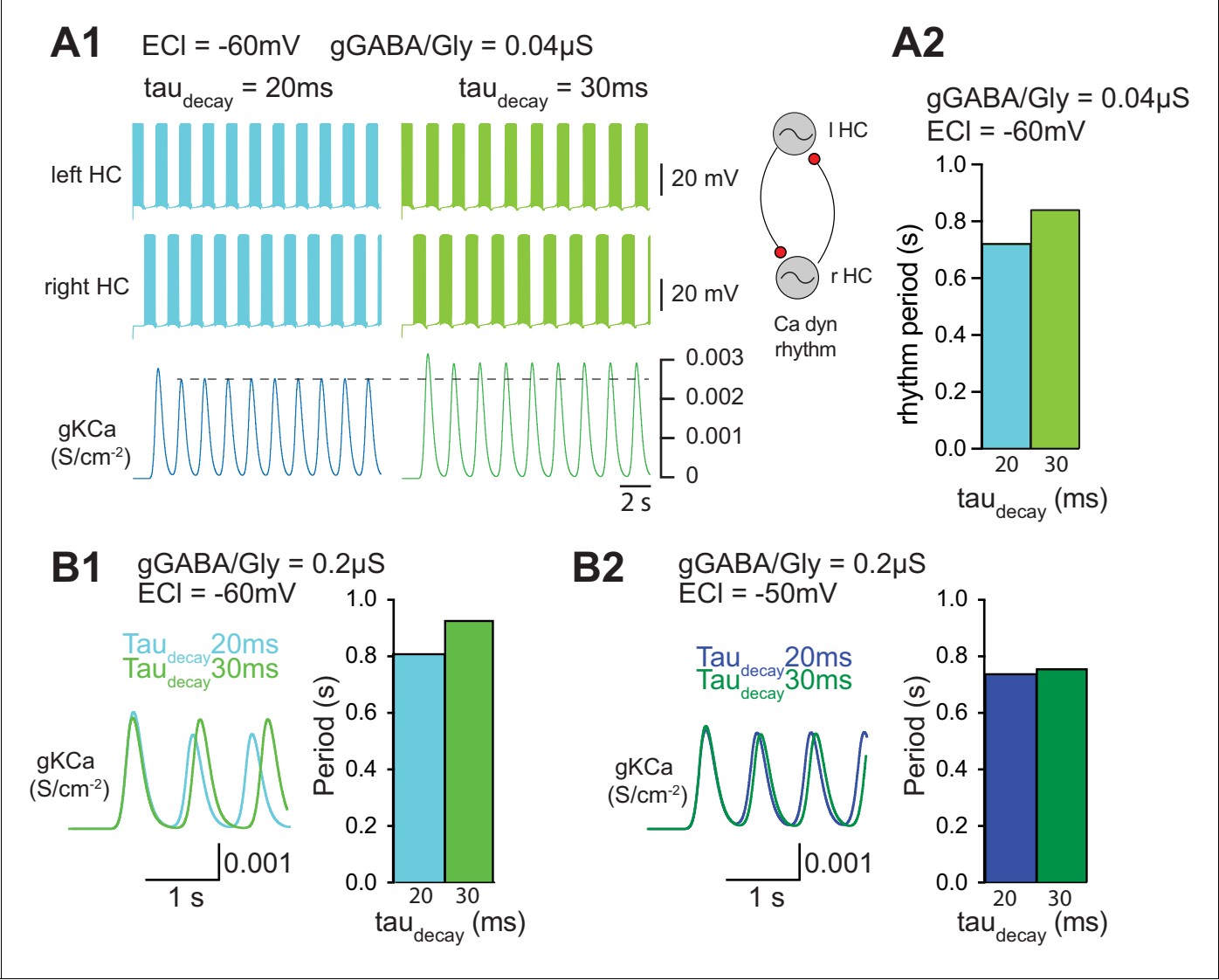

**Figure 7.** Functional consequence of increasing $tau_{decay}$ on locomotor activity. (**A**) Effect of $tau_{decay}$ on the locomotor rhythm when $E_{GABAAR}$ ($E_{Cl}$) is set to −60 mV (**A1-A2, B1**) and −50 mV (**B2**).

support the conclusion that fetal E17.5 SOD MNs are also hyperexcitable. The following will consider possible compensatory mechanisms that counteract the reduction in effective inhibitory input and the hyperexcitability of SOD MNs.

## The frequency and amplitude of inhibitory synaptic inputs are only weakly affected in fetal SOD MNs

In our study we show that frequency of inhibitory synaptic inputs is reduced in SOD MNs. This conclusion was supported by both physiological and anatomical observations. Firstly, SOD MNs displayed less frequent mIPSCs (~6%) (*Figure 4A2*), which correlates with the finding that VIAAT/synaptophysin co-localization was diminished (~23%) in these mice (*Figure 5B3*). Second, the amplitudes of the mIPSCs were slightly smaller in SOD1 MNs than in WT, especially for large events (*Figure 4A3*) and sIPSC events also exhibited this tendency (*Figure 4B3*). These main features of SOD IPSCs compared to WT are summarized in the schematic representation of *Figure 8A1–A2*.

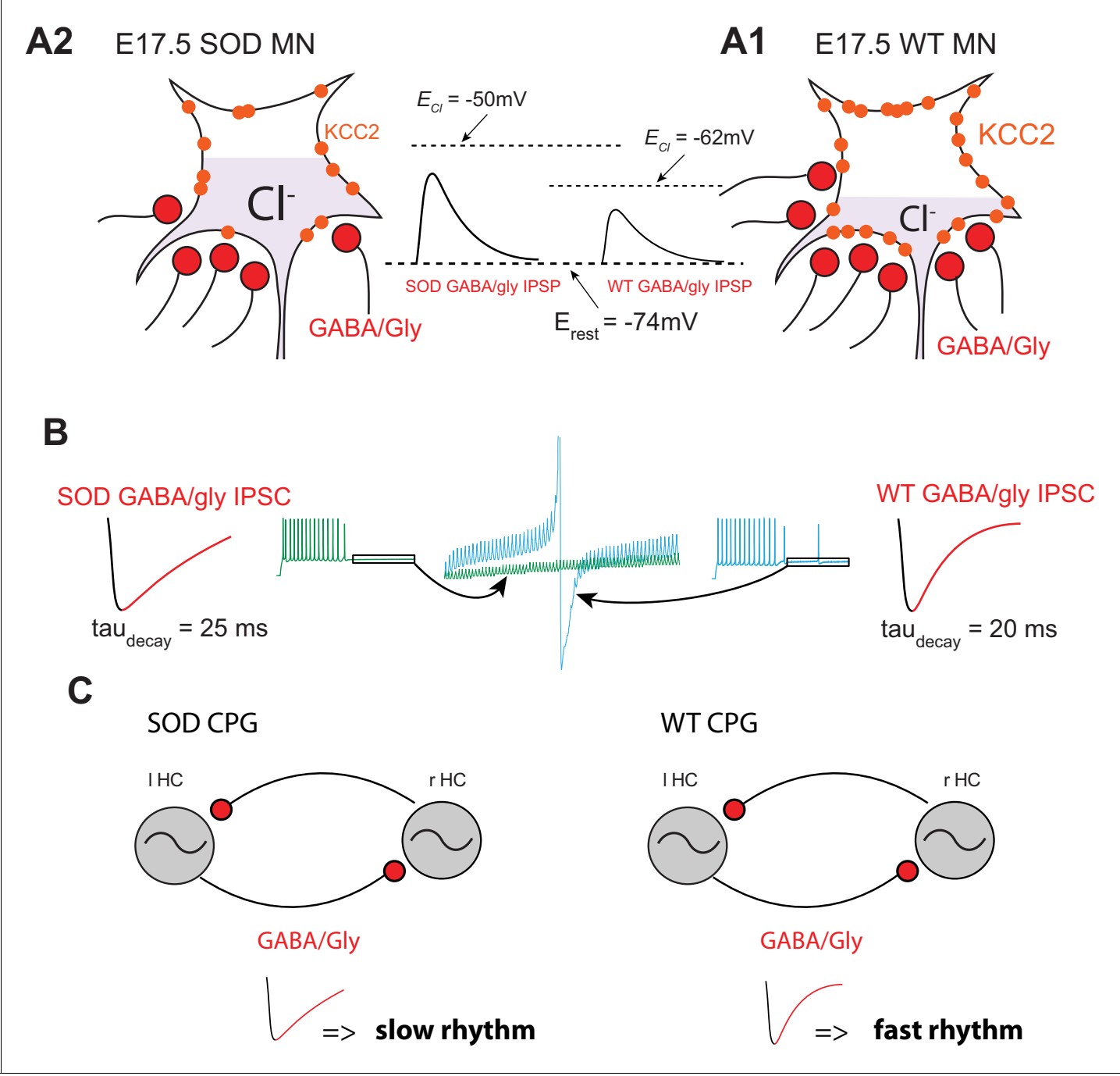

**Figure 8.** Summary. (A) Schematic drawing summarizing the altered inhibitory inputs (~25% reduction, see *Figure 3B3*) to fetal SOD MNs. [Cl⁻]ᵢ is higher in SOD MNs (A1) than in WT MNs (A2) because of a KCC2 down-regulation, leading to an increased GABA/Gly-induced depolarizing effect (see insets). (B) Consequence of increasing *tau_{decay}* on the GABA/gly inhibitory effect in SOD-like MNs. Due to an accumulation in the intracellular compartment, E_{GABAAR} exerts a strong depolarizing effect. A burst of spikes generated by MNs is hardly blocked by a barrage of GABA/gly events (see blue traces) when *tau_{decay}* is set to 20 ms. Increasing *tau_{decay}* to 25 ms allows a better summation of the shunting component of the depolarizing GABA/gly post-synaptic event leading to a better clamp of E_m towards E_{GABAAR} and to the blockade of MN discharge (see green traces). (C) Impact of a larger *tau_{decay}* in SOD MNs on the frequency of the locomotor rhythm.

The reduced degree of synaptic input to mouse prenatal SOD MNs assessed by VIAAT/synapto-physin staining is reminiscent of anatomical data from adult human ALS patients in which a reduction of synaptophysin was also identified (*Ikemoto et al., 2002*). It cannot be explained by a reduced SOD MN size (i.e., smaller cells can host fewer synapses) because we have shown in a previous article that soma perimeters and soma surface of WT MNs and SOD MNs are comparable (*Martin et al., 2013*). Even though the reduced degree of synaptic input might reflect a developmental delay in the maturation of SOD SCs compared to WT, data from the literature seems to indicate that this reduction persists with time. In fact, MNs cultured from other ALS patients were shown to progressively lose their synaptic activity (*Devlin et al., 2015*), as was also shown in pre-symptomatic (P60-P120) adult SOD mice (*Zang et al., 2005*).

The mixed GABA/Glycine IPSCs recorded in our study are likely to reflect a major contribution of GlyR because of the fast $tau_{decay}$ (*Gao et al., 2001*; *Muller et al., 2006*). Therefore, IPSCs, which were slightly reduced in amplitude in our recordings in SOD MNs, represent glycinergic events whereas GABAergic IPSCs amplitude remained unaffected. This hypothesis is supported by the fact that the ratio gGABA$_A$R/Cm measured in our perforated patch-clamp recordings did not differ between SOD and WT MNs (*Table 1*). Interestingly, a reduction in amplitude of glycinergic synaptic events but not GABAergic events was described in SOD1$^{G93A}$ lumbar MNs maintained in culture (*Chang and Martin, 2011*). Less frequent and smaller amplitude IPSCs convey a reduced inhibitory drive onto prenatal SOD1 MNs. Measurements of IPSC amplitudes derived from CsCl medium-based whole-cell recordings in which E$_{GABAAR}$ approached 0 mV. Therefore, WT and SOD MNs shared the same artificial driving force for chloride and so the observed reduction of IPSC amplitude in SOD MNs was likely to be related to a change in post-synaptic GlyR as described in cultured SOD1$^{G93A}$ MNs (*Chang and Martin, 2011*).

## A prolongation of inhibitory synaptic events in SOD MNs as a compensatory mechanism

Our data revealed a noticeable increase of $tau_{decay}$ in both SOD1 mIPSCs and sIPSCs. We found a mean $tau_{decay}$ of around 20–25 ms, which is in the same range as that previously described for mIPSCs during perinatal stages in rat lumbar MNs (*Gao et al., 1998*). Because a part of our study did not differentiate unequivocally between GABA$_A$R-mediated versus GlyR-mediated mIPSCs, we pharmacologically dissected GABA$_A$R-mediated versus GlyR-mediated mIPSCs. This confirmed that the mean $tau_{decay}$ is between $tau_{decay}$ of pure GABA$_A$R-mediated mIPSCs and pure GlyR-mediated mIPSCs and that both types of mIPSCs exhibit a longer relaxation. Interestingly, GlyRs switch from immature homomeric α2 GlyRs with slow decay kinetics of glycinergic mIPSCs to heteromeric GlyRs that include α1, α3 and β subunits with fast kinetics (*Legendre, 2001*; *Raltschev et al., 2016*). E17.5 SOD1 MNs may therefore exhibit a delayed development with a preponderance of homomeric α2 GlyRs. However, this is unlikely to be the case, since most electrical properties measured from birth to postnatal day 12 in SOD1$^{G93A}$ MNs instead show an accelerated rate of maturation (*Quinlan et al., 2011*). During perinatal development of rat spinal MNs, GlyR-mediated mIPSCs become dominating over GABA$_A$R-mediated mIPSCs (*Gao et al., 2001*). We also found a majority of pure GlyR-mediated mIPSCs and found the same ratio of pure GABA$_A$R-mediated mIPSCs and pure GlyR-mediated mIPSCs in SOD and WT MNs, indicating similar prenatal developmental stage for both genotypes.

Considering that SOD SCs do not express a development delay compared to WT SCs from the same littermate, we questioned the origin of the prolonged $tau_{decay}$ in GABA$_A$R- and GlyR-mediated mIPSCs, sIPSCs and eIPSCs. It has been demonstrated that high [Cl$^-$]$_i$ slows down the decay of glycine- and GABA-mediated inhibitory synaptic events (*Pitt et al., 2008*; *Houston et al., 2009*), reducing the firing probability of Purkinje neurons (*Houston et al., 2009*). Interestingly, the structural basis of this effect was described as a direct effect of chloride ions acting in the pore of glycine and GABA channels (*Moroni et al., 2011*). Therefore the elevated [Cl$^-$]$_i$ likely accounts for the prolonged relaxation of GABA/Gly IPSCs in SOD MNs. The increase in [Cl$^-$]$_i$ affect GABA$_A$R and GlyR and thus IPSCs duration (*Pitt et al., 2008*; *Houston et al., 2009*; *Moroni et al., 2011*). In P10 rat MNs a change from 10 mM [Cl$^-$]$_i$ to 131 mM led to an increase in $tau_{decay}$ from 9.2 ms to 19.8 ms (*Pitt et al., 2008*). In P12 Purkinje cells a change from 10 mM [Cl$^-$]$_i$ to 150 mM led to an increase in $tau_{decay}$ from 14 ms to 19.8 ms (*Houston et al., 2009*). Finally, in HEK-293 cells, a change from 10 mM [Cl$^-$]$_i$ to 30 mM and then 131 mM led to an increase in $tau_{decay}$ from 9.6 ms to 12 ms and then

22 ms (*Pitt et al., 2008*). Even though our data highlight a more prolonged $tau_{decay}$ in SOD GABA$_A$R/Gly IPSCs, we cannot directly link $tau_{decay}$ values to physiological [Cl$^-$]$_i$ values, because $tau_{decay}$ values have been collected from CsCl recordings in which [Cl$^-$]$_i$ was set as being elevated.

Impaired KCC2 expression or function is involved in a number of neurodevelopmental and neurological disorders (*Doyon et al., 2016*). We provide evidence that altered E$_{GABA}$ in E17.5 SOD1$^{G93A}$ mouse embryo likely arises from down-regulation of KCC2 in the lumbar cord and lumbar MNs. Again this down-regulation of KCC2 is unlikely linked to the decrease in the size of the SOD MNs, that is smaller neurons have less volume of the cytosol and less surface of the membrane to host KCC2, because soma perimeters and surface of E17.5 WT MNs and SOD MNs were found as similar (*Martin et al., 2013*). KCC2 protein has been quantified at the soma level and proximal dendrites that are not affected in SOD1 MNs. Also KCC2 efficacy and E$_{GABAR}$ have been assessed at the soma level. SOD MNs do not have less volume to adjust and the lower efficacy of KCC2 in SOD MNs is not related to a change in MN size.

In hippocampal pyramidal neurons, NKCC1 up-regulation has been the proposed mechanism that drives transient perinatal GABA switch in rodent models of autism (*Tyzio et al., 2014*). However, our data show that NKCC1 is unaltered in the E17.5 lumbar SCs. KCC2 down-regulation in the hippocampus delays postnatal GABA shift in oxytocin receptor knockout mouse model of autism (*Leonzino et al., 2016*). More recent evidence shows that KCC2 down-regulation explains long-lasting effects of adolescent nicotine exposure (*Thomas et al., 2018*). In ALS-vulnerable motoneurons of adult SOD1$^{G93A}$ mouse lumbar cord, a strong decrease of mRNA and protein expression levels was reported (*Fuchs et al., 2010*). Thus, KCC2 is a likely molecular substrate underlying perinatal alteration of chloride homeostasis in the lumbar spinal cord.

It may be surprising to observe a $tau_{decay}$ more prolonged in SOD MNs when [Cl$^-$]$_i$ is imposed (0 mV or −45 mV). Indeed, when we imposed similar [Cl$^-$]$_i$ in SOD and WT MNs, we expected a similar $tau_{decay}$ in SOD and WT. However, since our data clearly indicate a deficit in KCC2 efficacy in SOD MNs, the actual [Cl$^-$]$_i$ may have been counteracted in WT MNs by the KCC2 protein more effectively than in SOD MNs. Therefore, we observed a longer relaxation of GABA$_A$R-mediated/GlyR mediated mIPSCs, sIPSCs and VLF-evoked IPSCs in SOD MNs compared to WT.

Our first series of computer simulations demonstrated that increasing $tau_{decay}$ from 20 ms to 25 ms clearly potentiated the efficacy of GABA/glycine inhibition. With a $tau_{decay}$ of 20 ms, a 20 Hz barrage of inhibitory inputs totally blocks WT-like MN firing activity (12 Hz, i.e. locomotor firing frequency, with E$_{Cl}$ = -60mV). In contrast, SOD-like MNs (with E$_{Cl}$ = -50mV) required a ~ 90 Hz inhibitory barrage to suppress their firing. Increasing $tau_{decay}$ to 25 ms significantly reduced the threshold cutting frequency to ~70 Hz in SOD-like MNs. Those blocking frequencies are in agreement with GABA/glycinergic event frequencies encountered in lumbar spinal MNs during fictive locomotion (>100 Hz) (*Figure 6—figure supplement 2C, D*). Increasing $tau_{decay}$ from 20 ms to 25 ms also leads to a higher capacity of shunting effect summation (*Branchereau et al., 2016*). This is clearly visible in *Figure 8B* in which the increased shunting effect tends to maintain E$_m$ towards E$_{GABAAR}$ and prevents spike occurrence (compare green and blue traces).

Our second series of computer simulations demonstrate that increasing $tau_{decay}$ slowed-down the locomotor rhythm (*Figure 7*). This is in agreement with our findings that the pharmacologically-evoked locomotor rhythm in SOD SCs was slower than in WT SCs (*Figure 3B*). The effect of IPSCs $tau_{decay}$ on locomotor rhythm was only observed when E$_{GABAAR}$ was set to −60 mV in the interneurons of a central-pattern generator (CPG, half-centers) (*Figure 7*). To our knowledge, however, no E$_{GABAAR}$ and g$_{GABA/Gly}$ measurements have been obtained in the CPG interneurons of ALS mouse models. In their study of 2–3 week in vitro organotypic slice cultures of the spinal cord (from E12-E13), *Medelin et al. (2016)* suggested that the Cl$^-$ reversal potential is similar in SOD1$^{G93A}$ and WT ventral interneurons, around −60 mV. However, neither the excitatory *versus* inhibitory nature of the recorded interneurons, nor their relationship with CPG circuitry were reported in their study. Our computer model of locomotor rhythm was based on half-center interneurons without considering the output MNs. Therefore it may be argued that changes in IPSCs $tau_{decay}$ observed in spinal MNs has nothing to do with the spinal locomotor rhythm generator. However, a recent study clearly demonstrated that MNs participate to the rhythm generation in neonatal mouse (*Falgairolle et al., 2017*). Therefore the effect of $tau_{decay}$ increase observed in SOD MNs could therefore impinge the locomotor rhythm frequency.

# Materials and methods

## Key resources table

| Reagent type (species) or resource | Designation | Source or reference | Identifiers | Additional information |
|---|---|---|---|---|
| Strain (*Mus musculus*) | B6SJL-Tg (SOD1*G93A)1Gur/J | The Jackson Laboratory | RRID:IMSR_JAX:002726 | |
| Antibody | anti-KCC2 rabbit polyclonal | Millipore | Cat#07–432 RRID:AB_11213615 | 1:1000 (WB) 1:400 (IHC) |
| Antibody | anti-NKCC1 mouse monoclonal | Developmental Studies Hybridoma Bank | T4 | 1:400 (WB) |
| Antibody | anti-VIAAT rabbit polyclonal | Provided by B. Gasnier, Paris Descartes University | VIAAT | 1:1000 (IHC) |
| Antibody | anti-synaptophysin mouse monoclonal | Millipore Sigma | Clone SVP38 RRID:AB_2315393 | 1:500 (IHC) |
| Antibody | anti-FoxP1 rabbit polyclonal | Millipore Sigma | Cat#AB2277 RRID:AB_10631732 | 1:500 (IHC) |
| Antibody | goat anti-rabbit secondary antibody Alexa Fluor 546 | Invitrogen | Cat#A-11035 | 1:500 (IHC) |
| Antibody | goat anti-mouse secondary antibody Alexa Fluor 647 | Invitrogen | Cat#A-21235 RRID:AB_2535804 | 1:500 (IHC) |
| Antibody | Goat anti-Rabbit IgG-heavy and light chain cross-adsorbed Antibody HRP Conjugated | Euromedex | Cat#A120-201P | 1:20000 (WB) |
| Antibody | Immun-Star GAM-HRP Conjugate | Bio-Rad | Cat#1705047 | 1:20000 (WB) |
| Commercial assay, kit | Qproteome Mammalian Protein Prep Kit | Qiagen | Cat#37901 | |
| Chemical compound, drug | Phosphatase Inhibitor Cocktail 2 | Millipore Sigma | Cat#P5726 | |
| Commercial assay, kit | DC protein assay | Bio-Rad | Cat#5000113 Cat#5000114 Cat#5000115 | |
| Commercial assay, kit | Mini-PROTEAN TGX Stain-Free Gels | Bio-Rad | Cat#4568034 | |
| Chemical compound, drug | 10x Tris/Glycine/SDS | Bio-Rad | Cat#1610732 | |
| Commercial assay, kit | Precision Plus Protein All Blue Standards | Bio-Rad | Cat#1610373 | |
| Commercial assay, kit | Trans-Blot Turbo Transfer Pack, nitrocellulose | Bio-Rad | Cat#1704158 | |
| Chemical compound, drug | Clarity Western ECL, Substrate | Bio-Rad | Cat#1705060 | |
| Chemical compound, drug | Tetrodotoxin (citrate free) | Latoxan Laboratory, France | Cat#L8503 | 0.2 μM |
| Chemical compound, drug | Kynurenic acid | Millipore Sigma | Cat#K3375 | 4 mM |
| Chemical compound, drug | Tubocurarine hydrochloride pentahydrate | Millipore Sigma | Cat#2379 | 10 μM |
| Chemical compound, drug | Dihydro-β-erythroidine hydrobromide | Bio-techne, France | Cat#2349/10 | 5 μM |

*Continued on next page*

*Continued*

| Reagent type (species) or resource | Designation | Source or reference | Identifiers | Additional information |
|---|---|---|---|---|
| Chemical compound, drug | Strychnine Hemisulfate salt | Millipore Sigma | Cat#S7001 | 3 µM |
| Chemical compound, drug | (2R)-amino-5-phos phonovaleric acid (DL-AP5) | Bio-techne, France | Cat#0105/10 | 40 µM |
| Chemical compound, drug | 6-cyano-7-nitroquin oxaline-2,3-dione (CNQX) | Bio-techne, France | Cat#0190/10 | 20 µM |
| Chemical compound, drug | SR 95531 hydrobromide (GABAzine) | Bio-techne, France | Cat#1262/10 | 3 µM |
| Chemical compound, drug | bumetanide | Millipore Sigma | Cat#B3023 | 10 µM |
| Chemical compound, drug | VU0240551 | Bio-techne, France | Cat#3888/10 | 10 µM |
| Chemical compound, drug | Isoguvacine hydrochloride | Bio-techne, France | Cat#0235/100 | 50 µM |
| Chemical compound, drug | Dopamine hydrochloride | Millipore Sigma | Cat#H8502 | 100 µM |
| Chemical compound, drug | Serotonin hydrochloride (5-HT) | Millipore Sigma | Cat#H9523 | 10 µM |
| Chemical compound, drug | N-Methyl-D-aspartic acid (NMDA) | Millipore Sigma | Cat#M3262 | 10 µM |
| Chemical compound, drug | Gramicidin from Bacillus aneurinolyticus (Bacillus brevis) | Millipore Sigma | Cat#G5002 | 10–20 µgml$^{-1}$ |
| Software, algorithm | NEURON 7.3 | NEURON | RRID:SCR_005393 | |

## Ethical considerations and mouse model

All procedures were carried out in accordance with the local ethics committee of the University of Bordeaux and European Committee Council directives. All efforts were made to minimize animal suffering and reduce the number of animals used. B6SJLTgN(SOD1-G93A)/1Gur/J mice expressing the human G93A Cu/Zn superoxide dismutase (SOD1) mutation (Gly93→ Ala substitution) were obtained from the Jackson Laboratory (https://www.jax.org/strain/002726). Heterozygous B6SJL-TgN(SOD1-G93A)/1 Gur/J (named SOD in this report) were maintained by crossing heterozygous transgene-positive male mice with B6SJL F1 hybrid females (Janvier labs, France). Gestation in SOD lasted for ~18.5 days, embryonic day 0.5 (E0.5) being defined as the day after the mating night. Experiments were performed on E17.5 fetuses, that is collected one day before their birth.

## Dissection and isolation of the embryonic spinal cord (SC)

Pregnant mice were sacrificed by cervical dislocation. A laparotomy was performed and the fetuses were removed after cutting the uterine muscle. Fetuses were removed from their individual embryonic sacs and transferred into cooled artificial cerebrospinal fluid (aCSF) oxygenated with a 95% $O_2$ and 5% $CO_2$ mixture. The composition of the latter was in mM: 114.5 NaCl, 3 KCl, 2 $CaCl_2$-$2H_2O$, 1 $MgCl_2$-$6H_2O$, 25 $NaHCO_3$, 1 $NaH_2PO_4$-$H_2O$, 25 D-Glucose, pH 7.4 and osmolarity 307 mosmol/kg $H_2O$. Only fetuses that displayed active movements were chosen for experiments. The fetuses selected were then decapitated and the SC preparation was dissected out. Their tails were preserved for subsequent genotyping. An incision was performed on the ventral side of the lumbar cord (between the midline and ventral roots) at the level of lumbar 4–5 (L4-L5) ventral roots, which innervate the extensor muscles, in order to remove the meninges. This exposes the MNs and makes them accessible for the patch-clamp electrode (generally 1–2 MNs recorded in each SC). The SC preparation was placed in a recording chamber and continuously superfused (~1.5 mL·min$^{-1}$) with

oxygenated aCSF. All experiments were carried out at constant temperature (30°C). Experiments were performed blindly without knowing the genotype of the animals. Fetuses were genotyped at the Genotyping Platform at the Magendie Neurocentre (Bordeaux). The genotyping was performed by standard PCR from mice tail samples using established primers and a protocol as stated by the Jackson Laboratory.

## Protein extraction and western blotting (WB)

WB analysis was performed on whole E17.5 lumbar spinal cords. For protein extraction, 15 WT and 15 SOD mice, from four different litters, were used. Each of the 15 lumbar spinal cords were divided into three groups of five lumbar SCs, crushed (micro-potter, PP750ACD), and homogenized with mammalian lysis buffer (Qproteome MammalianProtein Prep Kit, Qiagen) with a phosphatase inhibitor cocktail (Millipore Sigma). Homogenate was incubated (5 min at 4°C), centrifuged (10 min at 4°C to 14,000 RCF) to recover the supernatant from debris and stored 48 hr at −80°C. Total protein concentration in the supernatant solution was determined by the DC protein assay (Bio-Rad) using an iMark microplate reader (Bio-Rad). Briefly, colorimetric technique was used to measure color change proportional to total protein concentration. Based on a determined concentration of total protein, volume of a supernatant solution that contains a desired amount of total protein could be calculated. Then, samples were deposited at −80°C for future use. WB was performed by denaturing the sample at 100°C for 5 min and loading 20 μg of protein of each sample (3 wells of WT and 3 wells of SOD samples) in Mini-PROTEAN TGX Stain-Free Gels (Bio-Rad) with migration buffer (2.5 mM Tris, 19.2 mM Glycine, 0.01% SDS prepared from 10X Tris/Glycine/SDS (Bio-Rad). Molecular weight markers (Precision Plus Protein All Blue Standards, Bio-Rad) were used in each individual gel. Electrophoresis of the samples and MW markers was powered by electrophoresis power supply EPS 600 (Pharmacia Biothe, Sweden) device at 250V. In order to be able to visualize the total proteins (stain free), the gel was activated in ChemiDoc MP imaging system (Bio-Rad). The transfer from gel to mini format 0.2 μm nitrocellulose membrane pack (Bio-Rad) was made using Trans-Blot Turbo Transfer System (Bio-Rad). Once the transfer was confirmed with ChemiDoc MP imaging system, membrane was blocked with 5% milk in tris-buffered saline (Millipore Sigma) containing 2% Tween-20 (Millipore Sigma) (TBST) buffer for 1 hr on agitator (Heidolph, UNIMAX 1010, Germany), and then incubated with primary antibody at 4°C overnight. The primary antibodies used were rabbit polyclonal anti-KCC2 (1:1000, reference Millipore 07–432, Millipore Sigma) or mouse monoclonal anti-NKCC1 (1:400, reference T4 from Developmental Studies Hybridoma Bank, The University of Iowa, USA). The next day, membrane was washed 3–5 times using TBST and incubated with goat anti-rabbit or anti-mouse Horseradish Peroxidase (HRP) conjugated secondary antibody diluted to 1:20000 (Euromedex or Bio-Rad) for 1.5 hr at room temperature. After 3–5 rinses in TBST, total proteins on the membrane were visualized using a ChemiDoc MP imaging system (Bio-Rad). Membrane was then incubated 5 min in Clarity western ECL substrate (Bio-Rad) and the KCC2 or NKCC1 protein was visualized using the using the ChemiDoc. Four rounds of WB were run from the same deposited WT and SOD spinal cord samples. The ~140 kDa KCC2 band and ~150 kDa NKCC1 were analyzed.

The stain-free staining of total proteins loaded was used as the normalization control to quantify the KCC2 or NKCC1 band. Quantification was performed using Image Lab (Bio-Rad). Data presented are normalized to mean values from WT samples.

## Immunohistochemistry

Immunohistochemistry was performed on frontal sections from lumbar SCs prepared as follows. The mice used were offspring of female Hb9-eGFP mice cross bred with SOD male mice. These mice express GFP in the dendrites and the soma of spinal MNs. The lumbar SC samples of SOD and WT embryos at E17.5 were fixed in 4% paraformaldehyde (PFA) for 2 hr at room temperature. They were rinsed three times with 0.1 M Phosphate Buffer Saline (PBS) and then cryoprotected in 15% sucrose for 24 hr, followed by 30% sucrose for another 24 hr. After placing them in Tissue-Tek (O.C. T. Compound, Sakura Finetek) and freezing, the samples were sliced using a Leica 3050 s Cryostat. Sections (20 μm thickness) were affixed on gelatinized slides and preserved at −25°C until their use. Each slide was rinsed three times with 0.1 M PBS, blocked with a medium containing 2% bovine serum albumin (BSA), then incubated for 48 hr with primary antibodies prepared in PBST (1% triton, BSA 0.2%). We processed the sections with an anti-synaptophysin antibody (1:500, mouse

monoclonal, Clone SVP38, Millipore Sigma) coupled either with a rabbit polyclonal antibody directed against the vesicular inhibitory amino acid transporter (1:1000, VIAAT, antibody Provided by B. Gasnier, Paris Descartes University). VIAAT reflects the synaptic release of the GABA and glycine neurotransmitters (*Dumoulin et al., 1999*). The rabbit polyclonal anti-FoxP1 antibody (1:500, AB2277, Millipore) was also used to assess the LMC identity of recorded MNs (*Figure 1—figure supplement 1*) (*Dasen et al., 2008*). For KCC2 staining, we used the rabbit polyclonal anti-KCC2 antibody (1:400, reference Millipore 07–432, Millipore Sigma). After three rinses, slides were incubated for 2 hr at room temperature with a goat anti-rabbit secondary antibody Alexa Fluor 546 (A-11035) and/ or goat anti-mouse secondary antibody Alexa Fluor 647 (A-21235) (1:500, Invitrogen, France) and then rinsed with 0.1 M PBS. After rinsing, slides mounted with an anti-fade reagent (Fluoromount, Electron Microscopy Sciences) and stored at 4°C in obscurity until confocal observation.

## Confocal microscopy

Samples were visualized either in the laboratory with a BX51 Olympus Fluoview 500 confocal microscope or in the Bordeaux Imaging Center (BIC) with a SP5 Leica. Serial optical sections (thickness 0.2 µm) were obtained using a x60 oil-immersion objective. Lasers were selected according to the wavelength required for visualization. The different transporters and proteins were visualized as spot aggregates. Spots were detected using the spot detector plugin in Icy version 1.8.1.0, a quantitative image analysis program (Institut Pasteur - CNRS UMR 3691). Spots were then quantified and the co-localization of synaptophysin with VIAAT was assessed in the marginal zone edging the MN soma location. The percentage of co-localization was calculated according to the surface of the synaptophysin positive area. The global and membrane KCC2 staining density was assessed using Image J. For the KCC2 membrane staining a specific macro, allowing to delineate the periphery of MNs and quantify the KCC2 staining density, was developed by Sébastien Marais from the BIC. Briefly, the contour of Hb9-eGFP MNs was manually outlined in order to build an area (five pixels on each side of the outline), in which KCC2 punctiform profiles were automatically detected. Density of KCC2 puncta was then calculated as the number of puncta relative to the outline area. Values are expressed as arbitrary unit (AU).

## Electrophysiological procedures and data analysis

Patch-clamp electrodes were constructed from thin-walled single filamented borosilicate glass (1.5 mm outer diameter, Harvard Apparatus, Les Ulis, France) using a two-stage vertical microelectrode puller (PP-830, Narishige, Tokyo, Japan). Patch electrode resistances ranged from 3 to 5 MΩ. All recordings were made with an Axon Multiclamp 700B amplifier (Molecular Devices, Sunnyvale, CA, USA). Data were low-pass filtered (2 kHz) and acquired at 20 kHz on a computer *via* an analog-to-digital converter (Digidata 1322A, Molecular Devices) and a data acquisition software (Clampex 10.3, Molecular Devices).

Motorized micromanipulators (Luigs and Neumann, Ratingen, Germany) were used to position the patch-clamp electrode on a visually identified MN using a CCD video camera (Zeiss Axiocam MR). Recorded MNs were located in the lateral column (*Figure 1—figure supplement 1*) and were identified by their pear-shaped cell bodies. Values of Rin and Cm also further confirmed the motoneuron identity of recorded neurons (*Martin et al., 2013*).

For the KCC2 efficacy assessment, the aCSF contained the following in mM: 127 NaCl, 6 KCl, 2 CaCl$_2$ 2H$_2$O, 1 MgCl$_2$ 6H$_2$O, 20 HEPES, 11 D-glucose, osmolarity 306 mosmol/kg H$_2$O, oxygenated with 100% O$_2$, pH adjusted to 7.4 with NaOH. The patch-clamp recording pipette was filled with the following in mM: 120 K-gluconate, 13 KCl, 1 CaCl$_2$ 2H$_2$O, 10 mM HEPES, 5 CsCl, 1 TEACl, 10 µM Verapamil, 2 ATP Mg$^{2+}$, 10 EGTA, pH = 7.4 adjusted with KOH. This led to a theoretical $E_{Cl}$ of −49 mV. In order to challenge the KCC2 efficacy, MNs were voltage-clamped at −20 mV (above $E_{Cl}$) while isoguvacine was applied during 40 s, inducing a massive influx of chloride ions. Actual $E_{Cl}$ was measured by holding the voltage from −100 mV to 10 mV using a 1 s ramp protocol (*Figure 2C1*). Subtracting the ramp current obtained in the presence of isoguvacine to the control ramp current allowed to determine $E_{Cl}$ (*Figure 2C1*).

For recordings of IPSCs, patch pipettes were filled with cesium chloride (CsCl) intracellular medium, the composition of which being (in mM): 130 CsCl, 4 MgCl$_2$-6H$_2$O, 10 HEPES, 10 EGTA, 4 ATP-2Na, 302 mosmol/kg H$_2$O. The pH was adjusted to 7.4 with the addition of CsOH. The

intracellular CsCl medium led to an $E_{GABAAR}$ that approached 0 (2.89 mV), allowing a clear visualization of Cl$^-$-dependent IPSCs. All synaptic events were recorded while holding the membrane potential of MNs at −70 mV.

In order to assess both Glu and GABA/Gly synaptic events in individual neurons (*Figure 6—figure supplement 2*), whole-cell patch-clamp recordings were conducted in voltage-clamp mode (holding membrane potential −45 mV). The intracellular medium was composed of (in mM): 130 KGluconate, 5 NaCl, 1 CaCl2, 2H2O, 10 HEPES, 10 EGTA, and 2 MgATP, (284 mosmol/kg H2O) adjusted to pH 7.4 using 1M KOH.

For $E_{GABAAR}$ assessment, gramicidin perforated patch-clamp recordings were conducted. For perforation, a gramicidin (Millipore Sigma) stock solution was dissolved at 10 µg ml$^{-1}$ in DMSO (Millipore Sigma) and diluted to a final concentration of 10–20 µgml$^{-1}$ in the intracellular medium composed of (in mM): 130 KCl, 10 HEPES, 10 EGTA, and 2 MgATP, (284 mosmol/kg H$_2$O) adjusted to pH 7.4 using 1M KOH. Gramicidin stock and diluted solutions were prepared <1 hr before each experiment. Pipette tips were filled with a filtered ~3 µL intracellular medium and subsequently back-filled with gramicidin solution. The GABA$_A$R specific agonist isoguvacine (50 µM, Bio-Techne, France) was pressure ejected (~ 3 psi; 50 ms) *via* a puff pipette placed in the vicinity of motoneuron somata with a PicoSpritzer II (Parker Hannifin Corporation, Fairfield, NJ, USA) driven by a programmable Master 8 Stimulator/Pulse Generator (Master-8, A.M.P.I. Jerusalem, Israel). Isoguvacine applications were repeated at different membrane voltages, allowing to assessment of the reversal potential of evoked GABA$_A$R-related currents. Measurements were corrected for liquid junction potentials (3.3 mV) calculated with the Clampex junction potential calculator. Using the Nernst equation, $E_{GABAAR}$ was calculated to be 1.5 mV. Thus, to confirm that the perforated-patch configuration had not ruptured during the experiment, repetitive $E_{GABAAR}$ tests were performed. Measurements leading to an $E_{GABAAR}$ close to 0 mV were considered as whole-cell recordings and cells were discarded. The Clampex membrane test was used to monitor the input resistance (Rin) and membrane capacitance (Cm): a −60 mV holding membrane potential and 5 mV steps (negative and positive, 40 ms duration) were chosen. The reversal potential for GABA$_A$R ($E_{GABAAR}$) was calculated by a linear fit (Prism seven from GraphPad Software Inc, USA). The *tau$_{rise}$* (ms), and *tau$_{decay}$* (ms) of IPSCs were determined using Spike 2 software version 7.15 (Cambridge Electronic Design, Cambridge, England) and by fitting a single exponential.

In order to elicit evoked IPSCs (eIPSCs) from MNs, a concentric bipolar wire stimulating electrode (FHC, USA) was placed on the ventro-lateral funiculus (VLF) in order to activate local inhibitory fibers. Single stimulations delivered to the VLF (duration 1 ms, intensity ranging 10–20 µA) were driven by a programmable Master 8 Stimulator/Pulse generator (Master-8, A.M.P.I. Jerusalem, Israel).

For extracellular recordings of locomotor-like activity, left and right lumbar ventral roots (at the level of L3 to L5) were recorded. Raw signals were collected with a high-gain AC amplifier (ISO-DAM8A-4 Bio-amplifier System, World Precision Instruments Ltd, Stevenage, UK). Filtered (cutoff frequency: 0.3–3 kHz) raw signals were integrated off-line and analyzed using Spike 2 software. Locomotor-like activity was elicited by applying a cocktail of 5-HT 10 µM, NMDA 10 µM and DA (100 µM) (5-HT/NMDA/DA). This cocktail has been previously established (*Han et al., 2007*; *Milan et al., 2014*) to evoke stable locomotor-like activity. Two 50 s representative episodes of strongly alternating locomotor-like activity (assessed by the Rayleigh's test) were analyzed in each WT and SOD experiment.

## Pharmacology

In gramicidin perforated experiments and in experiments performed for assessing the KCC2 efficacy, GABA$_A$R responses were isolated by using a cocktail of drugs containing 0.2 µM tetrodotoxin (TTX, Latoxan Laboratory, France), 4 mM kynurenic acid (Millipore Sigma), 10 µM (+)-tubocurarine (Millipore Sigma), 5 µM Dihydro-β-erythroidine hydrobromide (DHβE, Bio-techne, France), and 3 µm strychnine (Millipore Sigma) that respectively blocked voltage-dependent Na$^+$ action potentials, and glutamate, cholinergic, and glycinergic input to MNs. In CsCl experiments, IPSCs were isolated pharmacologically using DL-AP5 40 µM ((2R)-amino-5-phosphonovaleric acid, Bio-techne, France) and CNQX 20 µM (6-cyano-7-nitroquinoxaline-2,3-dione, Bio-techne, France). mIPSCs were isolated in the presence of 0.2 µM TTX (Latoxan, France). GABA and glycine mIPSCs were isolated by adding 3 µM strychnine or 3 µM GABAzine (SR 95531 hydrobromide, Bio-techne, France), respectively. These blockers were added to the cocktail containing 0.2 µM TTX, 4 mM kynurenic acid, 10 µM (+)-

tubocurarine and 5 µM DHβE. In experiments assessing the KCC2 efficacy, 10 µM bumetanide (Millipore Sigma) was applied to block NKCC1 and 10 µM VU0240551 (Bio-techne, France) (10 µM) to block KCC2. Serotonin (5-HT, 10 µM), dopamine (DA, 100 µM) and N-Methyl-D-aspartic acid (NMDA, 10 µM) were from Millipore Sigma.

## Membrane properties and data analysis

Data analysis was performed using Clampfit 10.6 (Axon Instruments), MiniAnalysis 6.0. or Spike 2. For synaptic event analysis, 100–200 s recording samples were selected from each MN. All selected synaptic events were concatenated for SOD *versus* WT MNs. After analysis, the information obtained included mean values for event amplitude (pA), $tau_{rise}$ (ms), $tau_{decay}$ (ms) as well as inter-event interval (IEI) (ms). The MN capacitance (Cm), membrane input resistance (Rin) and access resistance (Ra) were recorded immediately after establishing whole-cell patch-clamp. The membrane voltage was held at −70 mV as this corresponds to the resting membrane potential of mice MNs at E17.5 (*Delpy et al., 2008*). As a result, all events appeared as downward events.

## Computer simulations

GABA/glycine synaptic events in E17.5 MNs were simulated using a multi-compartment neuron model that was elaborated with the NEURON 7.3 program (*Hines and Carnevale, 1997*). Two simulated neurons were constructed: an E17.5 WT-like MN and an E17.5 SOD-like MN (*Figure 6—figure supplement 1*). These canonical MNs, designed from the average topology of real E17.5 MNs, were built on models used in a previous study (*Martin et al., 2013*). Both MNs were virtually identical, with similar channels and morphologies, except for the terminal dendritic segments of the SOD-like MN that were 40% less in extent than the WT terminal segments (*Martin et al., 2013*). In each section (dendrites and axon), the number of segments was an odd number calculated according to the d-lambda rule (*Hines and Carnevale, 2001*; *Carnevale and Hines, 2006*). The following conductances were used: a passive leakage current (*Ileak*), a transient K channel ($I_A$), a voltage-dependent calcium-activated potassium channel ($I_C$) also called $I_{K(Ca)}$ (*McLarnon et al., 1995*; *Gao and Ziskind-Conhaim, 1998*) and a high-threshold calcium current ($I_L$) (*Walton and Fulton, 1986*). *Ileak* was simulated in each MN section (*Ileak = (Eleak-E) • Gleak*, with *Eleak* = −73 mV and *Gleak* = 1/*Rm*). *Rm*, the specific membrane resistance was set to 21200 Ω.cm$^2$, in order to obtain an input resistance of 120 MΩ in the WT MN. $I_A$ was present in the axon initial segment (AIS), with $G_{IA}$ = 0.0033 S.cm$^{-2}$; $I_C$ was present in soma, with $G_{IC}$ = 0.0025 S.cm$^{-2}$. $I_L$ was present in the soma, with $G_{IL}$ = 8.10$^{-5}$ S.cm$^{-2}$. Each of the axon segments (n = 750) was equipped with Na and K Hodgkin Huxley (HH) channels (GNa = 0.012 S.cm$^{-2}$ and GK = 0.0036 S.cm$^{-2}$, respectively) used to generate spikes. The density of Na and K channels in the initial segment (GNa = 0.5 S.cm$^{-2}$ and GK = 0.15 S.cm$^{-2}$) were adjusted in order to obtain a spike threshold of −48 mV for the WT MN (for more details see Appendix 1).

In addition, calcium dynamics (*Blaustein, 1988*) were added in the soma to reproduce calcium accumulation, diffusion and pumping (for more details see [*Destexhe et al., 1993*]). The parameters used in the intracellular calcium dynamics were: *depth* = 0.17 µm, $tau_r$ = 1e$^{10}$ ms, $Ca_{inf}$ = 0.0002 mM, *Kt* = 0.00025 mM.ms-1, *Kd* = 0.0001 mM, $Ca_i$ = 2.4e$^{-4}$ mM and $Ca_o$ = 3 mM.

Inhibitory and excitatory synaptic inputs were inserted on the somatic MN compartment using two exponentials equations (one for rising phase and the other for decay phase). Depolarizing GABAergic/glycinergic post-synaptic current kinetics was set with $tau_{rise}$ = 0.3 ms and $tau_{decay}$ = 15/20 ms (for WT/SOD, respectively). Excitatory synaptic current kinetics was set with $tau_{rise}$ = 0.1 ms and $tau_{decay}$ = 10 ms. Equations concerning channel properties and GABA$_A$R synaptic activation are summarized in the Appendix 1.

## Statistical analysis

GraphPad Prism 7 software was used to analyze all the data. The results are presented as means ± the standard error of the mean (SEM) unless otherwise specified. *n* is the number of MNs used in the analysis and *N* the number of fetuses. Significance was determined as p<0.05 (*), p<0.01 (**), p<0.001 (***) or p<0.001 (***). The difference between the cumulative frequencies was analyzed using the Kolmogorov-Smirnov (K-S) test. The statistical differences between two data sets were assessed with the Mann Whitney test for non-parametric data.

## Acknowledgements

We thank Marie-Paule Algeo, Marie-Alix Derieppe and Nathalie Argenta for excellent technical assistance and animal breeding. A part of the microscopy was done in the Bordeaux Imaging Center, a service unit of the CNRS-INSERM and Bordeaux University, member of the national infrastructure France BioImaging. The help of Sébastien Marais is specifically acknowledged. This work was supported by funding from the 'Fédération pour la Recherche sur le Cerveau' (FRC) and the 'Association pour la recherche sur la Sclérose Latérale Amyotrophique et autres maladies du Motoneurone' (ARSLA).

## Additional information

### Funding

| Funder | Author |
| --- | --- |
| Fédération pour la Recherche sur le Cerveau | Pascal Branchereau |
| Association pour la Recherche sur la Sclérose Latérale Amyotrophique et autres Maladies du Motoneurone | Pascal Branchereau |

The funders had no role in study design, data collection and interpretation, or the decision to submit the work for publication

### Author contributions

Pascal Branchereau, Conceptualization, Data curation, Formal analysis, Supervision, Funding acquisition, Validation, Investigation, Visualization, Methodology, Project administration; Elodie Martin, William Cazenave, Fara Hodeib, Amandine Laupénie, Urvashi Dalvi, Hongmei Zhu, Formal analysis, Investigation; Anne-Emilie Allain, Conceptualization, Formal analysis, Investigation; Laura Supiot, Data curation, Formal analysis; Daniel Cattaert, Conceptualization, Data curation, Software, Formal analysis, Supervision, Validation, Investigation, Visualization, Methodology

### Author ORCIDs

Pascal Branchereau https://orcid.org/0000-0003-3972-8229

### Ethics

Animal experimentation: All procedures were carried out in accordance with the local ethics committee of the University of Bordeaux (Saisine SOD1G093A - APAFiS #19366) and European Committee Council directives. All efforts were made to minimize animal suffering and reduce the number of animals used.

### Decision letter and Author response

Decision letter https://doi.org/10.7554/eLife.51402.sa1
Author response https://doi.org/10.7554/eLife.51402.sa2

## Additional files

### Supplementary files

• Transparent reporting formSign-off

### Data availability

All data generated or analysed during this study are included in the manuscript and supporting files.

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

## Appendix 1

### MN simulations

Simulations were carried out with NEURON 7.3 (*Carnevale and Hines, 2006*). A WT- and a SOD1[G93A]–like MN were used (*Martin et al., 2013*). Both MNs were identical, with similar channel properties and morphologies, except for the terminal dendritic segments of the SOD1[G93A]–like MN that were 40% less extensive than those of the WT. The WT-like canonical E17.5 MN was composed of a cell body (11 µm in diameter), 15 main dendrites, and an initial segment and axon. The initial segment (length = 30 µm) was separated from the soma by a passive compartment (length = 10 µm). Since we sought to study fine consequences of IPSc shape on MN excitability, the axon length was increased to 2000 µm in order to avoid seal-end effects. In each section (dendrite and axon), the number of segments was an odd number calculated according to the d-lambda rule (*Hines and Carnevale, 2001*; *Carnevale and Hines, 2006*). The properties of each compartment could be defined independently.

### Passive properties of all compartments

In each compartment, the capacitance $c_m$ (µF) was calculated in accordance with the equation:

$$c_m = C_m \times area$$

(*area* is the membrane surface of the compartment in cm$^2$, and $C_m$ is the specific membrane capacitance, set to 1 µF.cm$^{-2}$).

In addition, a passive leak current was simulated in each compartment by the equation:

$$I_{leak} = (E_{leak} - E) \times G_{leak}$$

($E_{leak}$ and $G_{leak}$ are the leak equilibrium potential and leak conductance, respectively. In the present simulations, $E_{leak}$ = −73 mV, and $G_{Leak}$ = 1/$R_m$; with $R_m$, the specific membrane resistance was set to 21200 Ω.cm$^2$, in order to obtain an input resistance of 120 MΩ in the WT-like MN (*Martin et al., 2013*). $E$ is the membrane potential).

All computations were carried out assuming a specific axoplasmic resistance, $R_a$, of 100 Ω.cm.

### Active properties of axon and initial segment

In addition to passive properties, each of the axon compartments possessed active properties simulated by Hodgkin and Huxley (HH) Na and K channels. Their densities, adjusted in order to obtain a spike threshold of −48 mV for the WT-like MN, were respectively:

$gNa_{hh}Max$ = 0.50 S.cm$^{-2}$ and $gK_{hh}Max$ = 0.15 S.cm$^{-2}$ in the initial segment

$gNa_{hh}Max$ = 0.012 S.cm$^{-2}$ and $gK_{hh}Max$ = 0.0036 S.cm$^{-2}$ in the axon. The formalism of HH channels were described by standard HH Na and K channels kinetic equations:

$$\mathrm{For\,Na\,channels}: gNa_{hh} = gNa_{hh}Max \times m \times m \times m \times h$$

$$\mathrm{Activation}: m' = \alpha_m \times (1-m) - \beta_m \times m$$

$$\alpha_m = A_{\alpha m} \times (k_{\alpha m} \times (E - d_{\alpha m}))/(1 - exp(-k_{\alpha m} \times (E - d_{\alpha m})));$$
$$A_{\alpha m} = 1\,(\mathrm{ms}^{-1}); k_{\alpha m} = 0.1\,(\mathrm{mV}^{-1}); d_{\alpha m} = -40\,(\mathrm{mV})$$

$$\beta_m = A_{\beta m} \times exp(k_{\beta m} \times (E - d_{\beta m})); A_{\beta m} = 4\,(ms^{-1}); k_{\beta m} = -0.55556\,(\mathrm{mV}^{-1}); d_{\beta m} = -65\,(\mathrm{mV})$$

$$\mathrm{Inactivation}: h' = \alpha_h \times (1-h) - \beta_h \times h$$

$$\alpha_h = A_{\alpha h} \times exp(k_{\alpha h} \times (E - d_{\alpha h})); A_{\alpha h} = 0.07\,(\mathrm{ms}^{-1}); k_{\alpha h} = -0.05\,(\mathrm{mV}^{-1}); d_{\alpha h} = -65\,(\mathrm{mV})$$

$$\beta_h = A_{\beta h}/(1 + exp(k_{\beta h} \times (E - d_{\beta h})); A_{\beta h} = 1(\text{ms}^{-1}); k_{\beta h} = -0.1(\text{mV}^{-1}); d_{\beta h} = -35(\text{mV})$$

$$\text{For K channels}: gK_{hh} = gK_{hh}Max \times n \times n \times n \times h$$

$$\text{Activation}: n' = \alpha_n \times (1 - n) - \beta_n \times n$$

$$\alpha_n = A_{\alpha n} \times (k \times (E - d_{\alpha n}))/(1 - exp(-k_{\alpha n} \times (E - d_{\alpha n})));$$
$$A_{\alpha n} = 0.1(\text{ms}^{-1}); k_{\alpha n} = -0.1(\text{mV}^{-1}); d_{\alpha n} = -55(\text{mV})$$

$$\beta_n = A_{\beta n} \times exp(k \times (E - d_{\beta n})); A_{\beta n} = 0.125(\text{ms}^{-1}); k_{\beta n} = -0.0125(\text{mV}^{-1}); d_{\beta n} = -65(\text{mV})$$

## Adaptation of the firing frequency by a somatic Ca-dependent K channel

In order to mimic the adaptation of firing frequency observed in real WT MNs, Ca-dependent K channels ($K_{Ca}$), L-type calcium channels and intracellular calcium dynamics were added in the soma.

The calcium dynamics was used to set the decay of the intracellular concentration of calcium based on a calcium pump (for more details see [*Destexhe et al., 1993*]) and first-order decay buffering:

$$dCa_i/dt = (Ca_{inf} - Ca_i)/tau_r$$

where $Ca_{inf}$ is the equilibrium intracellular calcium value (usually in the range of 200–300 nM) and $tau_r$ is the time constant of calcium removal. The dynamics of submembrane calcium is usually thought to be relatively fast, in the 1–10 millisecond range (*Blaustein, 1988*).

$I_A$ is present in the axon initial segment (AIS), with $G_{IA} = 0.0033\ S.cm^{-2}$; $I_C$ is present in soma, with $G_{IC} = 0.0025\ S.cm^{-2}$. $I_L$ is present in the soma, with $G_{IL} = 8.10^{-5}\ S.cm^{-2}$.

The parameters used for the intracellular calcium dynamics were:

*depth* = 0.17 μm, $tau_r$ = 1e10 ms, $Ca_{inf}$ = 0.0002 mM, Kt = 0.00025 mM.ms-1, Kd = 0.0001 mM, $Ca_i$ = 2.4e$^{-4}$ mM and $Ca_o$ = 3 mM.

## CPG SIMULATIONS

Simulations of CPG activity were achieved with NEURON 7.3 using two pacemaker interneurons coupled *via* reciprocal inhibition. The interneurons were composed of a cell body (diam. = 10 μm), a main dendrite (length = 200 μm, diam. = 5 μm), an initial segment and an axon. The initial segment (length = 20 μm) and the axon (length = 1000 μm) had the same diameter (2 μm). The number of segments was an odd number calculated according to the d-lambda rule.

### Passive properties of all compartments

Based on the same formalism as in MN, parameters values were adjusted as follow for the two pacemaker interneurons. In each compartment, the specific capacitance $C_m$ was set to (1 μF.cm$^{-2}$), $R_m$, the specific membrane resistance was set to 80000 Ω.cm$^2$, and $R_a$, the specific axoplasmic resistance, was set to 100 Ω.cm. Resting membrane potential was set to −75 mV by adjusting $E_{Leak}$.

### Active properties of axon and initial segment

In addition to passive properties, the soma and each of the axon compartments possessed active properties simulated by Hodgkin and Huxley (HH) Na and K channels. The formalism of HH channels were identical to that used for MNs (see above). Their densities were adjusted to fit a somatic spike amplitude of 105 mV, and general properties of pacemaker interneurones described in *Tazerart et al. (2008)*. They were respectively:

$gNa_{hh}Max$ = 0.020 S.cm$^{-2}$ and $gK_{hh}Max$ = 0.005 S.cm$^{-2}$ in the soma

$gNa_{hh}Max$ = 0.400 S.cm$^{-2}$ and $gK_{hh}Max$ = 0.075 S.cm$^{-2}$ in the initial segment

$gNa_{hh}Max$ = 0.024 S.cm$^{-2}$ and $gK_{hh}Max$ = 0.0036 S.cm$^{-2}$ in the axon.

Small densities of HH K channels were also inserted into the dendritic compartment ($gK_{hh}Max$ = 0.0008 S.cm$^{-2}$) to prevent inactivation of HH Na channels present in the soma.

Pacemaker activities of the two interneurons were obtained using a persistent sodium channel (NaP) and a calcium-dependent potassium channel ($K_{Ca}$), which were inserted into the soma and dendrite.

$gNaPMax$ = 0.02 S.cm$^{-2}$ in soma and $gNaPMax$ = 0.06 S.cm$^{-2}$ in the dendrite.

$gKCaMax$ = 0.2 S.cm$^{-2}$ in soma and $gKCaMax$ = 0 S.cm$^{-2}$ in the dendrite.

KCa kinetics was modeled according to *Stacey and Durand (2000)*.

To ensure calcium entry to activate KCa, $I_L$ was present in the soma, with $G_{IL}$ = *0.001 S.cm$^{-2}$*. Calcium dynamics was also used as in MNs, with the following settings:

*depth* = 0.1 µm, *tau$_r$* = 5e$^5$ ms, *Ca$_{inf}$* = 1e$^{-6}$ mM, *Kt* = 1e$^{-6}$mM.ms-1, *Kd* = 1e$^{-7}$mM, *Ca$_i$* = 1e$^{-5}$mM and *Ca$_o$* = 0.06 mM.

