## [Decision Letter]

**Acceptance summary:**

This work uncovered cellular mechanisms leading to impaired inhibitory synaptic transmission and subsequent hyperexcitability in motor neurons during embryonic development in SOD193A mice, a mouse model of amyotrophic lateral sclerosis. Furthermore, it demonstrated that the coordinated locomotor activity was preserved during postnatal development due to compensatory increase in the decay of inhibitory postsynaptic currents.

**Decision letter after peer review:**

Thank you for submitting your article "Relaxation of synaptic inhibitory events as a compensatory mechanism in fetal SOD spinal motor networks" for consideration by *eLife*. Your article has been reviewed by two peer reviewers, and the evaluation has been overseen by a Reviewing Editor and Eve Marder as the Senior Editor. The following individual involved in review of your submission has agreed to reveal their identity: Laura Ballerini (Reviewer #2).

The reviewers have discussed the reviews with one another and the Reviewing Editor has drafted this decision to help you prepare a revised submission.

Summary:

The work by Branchereau et al. addresses a relevant and timely placed topic, related to the general issue of the presence of pre-symptomatic alterations in immature neuronal networks affecting synaptic signalling in familial ALS. In particular, spinal synaptic inhibition efficacy in mutated ALS mice has been recently indicated as potentially altered in early pre-motor network function during spinal maturation in SOD1G93A model mice. The work is nicely designed, written and conducted via sophisticated experimental tools. The core of the work is the hyper-excitability of spinal MN due to a weakening of inhibition which is proposed to be linked to a prolonged immature state, or delayed maturation of KCC2 expression in MN, leading to a more depolarized EGABAAR. The authors hypothesized that the development of inhibition is delayed in SOD1G93A MN. While abnormal spinal inhibition maturation has been previously suggested (Change and Martin, 2009, 2011, Medelin et al., 2016), the current work proposes the specific involvement of chloride transporters in this process. While the results are novel and exciting, multiple alternative mechanisms should be considered and experimentally tested to strengthen the conclusions of the manuscript.

Essential revisions:

1) Many of changes reported could be explained by the change in neuronal morphology – the authors should report these morphological changes at the developmental stage they perform the recordings. For example, decreased capacitance and membrane conductance may be explained by smaller size of SOD mouse motor neurons. Spontaneous IPSCs were less frequent in SOD animals, which correlated with a reduction of synaptic VIAAT44 positive terminals. This finding can be explained as well by the reduced cell size: smaller cells can host fewer synapses. The authors, however, do not discuss such possibility and do not provide any other mechanism for a reduced number of terminals. The authors also suggest that depolarizing shift in EGABA is due to a decrease in the level of KCC2. However, reduced KCC2 protein can also be linked to the decrease in the size of the SOD neurons: smaller neurons have less volume of the cytosol and less surface of the membrane to host KCC2. The levels of KCC2 and NKCC1 should be normalized to the volume fraction of SOD motor neurons. In fact, depolarizing GABA is linked to delayed developmental downregulation NKCC1.

2) The development of spinal inhibition is complex and involves 3 major events: changes in ECl; a functional switch between GABA and glycine inhibition; changes in GABAA and glycine receptor subunits composition. The current Results and Discussion should consider these three events, all affecting physiologically IPSC duration, if the framework of neurodevelopmental disease is to be strengthened for familial SOD1 ALS disease. The authors should pharmacologically dissect synaptic currents mediated by GABA and glycine receptors. The GABA/Gly receptor switch is an important player in shaping synaptic inhibition duration, thus identifying the diverse event and their relative contribution is important.

3) The increase in intracellular chloride (convincingly shown in this work) will affect GABAA and Gly receptors and thus the event duration (Pitt et al., 2008; Houston et al., 2009; Moroni et al., 2011), however the reported change (from 12 mM to 19 mM) is matching the change in decay detected in mIPSCs (from 16 ms to 20 ms)? In addition, other mechnanisms may explain the slower decay time of IPSCs in SOD motor neurons. The GABAA receptors are highly permeable to HCO_3_^-^. The different driving force for Cl^-^ and HCO_3_^-^ in WT and SOD motor neurons may be responsible for the difference in the IPSC decay time.

4) It is also not clear why the authors are indicating the prolonged relaxation of synaptic inhibitory events as a "compensation" (title, Abstract etc etc) since this is simply due to the higher intracellular concentration of Cl−, that strongly influence the duration of GABAergic and glycinergic synaptic inhibition while affecting the reverse potential. How are the average mIPSC (recorded events) charges when considering the reduced amplitudes?

5) The authors report an exciting finding that EGABA is 10 mV more depolarizing in SOD motor neurons. Since motor neurons express tonic GABAA conductance, this conductance can make cells more excitable and promote their spontaneous firing. A similar phenomenon has been previously reported in hippocampal interneurons. The authors may consider the depolarizing tonic GABAA conductance in SOD motor neurons.

---

## [Author Response]

Essential revisions:1) Many of changes reported could be explained by the change in neuronal morphology – the authors should report these morphological changes at the developmental stage they perform the recordings. For example, decreased capacitance and membrane conductance may be explained by smaller size of SOD mouse motor neurons.

This point is now included in the Discussion section. We have already addressed this question in a previous paper, in which we showed that soma perimeters and soma surface of WT MNs and SOD MNs are comparable (Martin et al., 2013).

Spontaneous IPSCs were less frequent in SOD animals, which correlated with a reduction of synaptic VIAAT44 positive terminals. This finding can be explained as well by the reduced cell size: smaller cells can host fewer synapses.

See Discussion section.

The authors, however, do not discuss such possibility and do not provide any other mechanism for a reduced number of terminals.

We have also discussed the possibility of a developmental delay.

The authors also suggest that depolarizing shift in EGABA is due to a decrease in the level of KCC2. However, reduced KCC2 protein can also be linked to the decrease in the size of the SOD neurons: smaller neurons have less volume of the cytosol and less surface of the membrane to host KCC2. The levels of KCC2 and NKCC1 should be normalized to the volume fraction of SOD motor neurons. In fact, depolarizing GABA is linked to delayed developmental downregulation NKCC1.

Since MN cell bodies size is similar in SOD1 and WT MNs (Martin et al., 2013), we did not normalized the level of KCC2 and NKCC1 to the volume fraction of SOD MNs.

Small MNs have less volume to adjust àKCC2 is more efficient in small MN.

This is a relevant remark but this is not the case. In fact, we have shown that soma perimeters and soma surface of WT MNs and SOD MNs are comparable (Martin et al., 2013). KCC2 protein has been quantified at the soma level and proximal dendrites that are not affected in SOD1 MNs. Again, KCC2 efficacy and E_GABAR_ have been assessed at the soma level.

2) The development of spinal inhibition is complex and involves 3 major events: changes in ECl; a functional switch between GABA and glycine inhibition; changes in GABAA and glycine receptor subunits composition. The current results and discussion should consider these three events, all affecting physiologically IPSC duration, if the framework of neurodevelopmental disease is to be strengthened for familial SOD1 ALS disease. The authors should pharmacologically dissect synaptic currents mediated by GABA and glycine receptors. The GABA/Gly receptor switch is an important player in shaping synaptic inhibition duration, thus identifying the diverse event and their relative contribution is important.

We have performed supplementary experiments in which, as suggested by the reviewer, pure GABA and Glycine mIPSCs were analyzed. Our new data are now included in Figure 4 that has been implemented to include the% of pure GABA mIPSCs and pure Glycine mIPSCs. New Figure 4 also includes the comparison between tau_decay_ calculated for pure GABA and Glycine mIPSCs.

3) The increase in intracellular chloride (convincingly shown in this work) will affect GABAA and Gly receptors and thus the event duration (Pitt et al., 2008; Houston et al., 2009; Moroni et al., 2011), however the reported change (from 12 mM to 19 mM) is matching the change in decay detected in mIPSCs (from 16 ms to 20 ms)?

This is an interesting comment. We have now included data from these authors in the

Discussion section. However, even though our data highlight a more prolonged tau_decay_ in SOD GABA_A_R/Gly IPSCs, we cannot directly link tau_decay_ values to physiological [Cl^-^]I values, because tau_decay_ values have been collected from CsCl recordings in which [Cl^-^]_i_ was set as being elevated.

In addition, other mechnanisms may explain the slower decay time of IPSCs in SOD motor neurons. The GABAA receptors are highly permeable to HCO_3_^-^. The different driving force for Cl^-^ and HCO_3_^-^ in WT and SOD motor neurons may be responsible for the difference in the IPSC decay time.

The ECl may be considered as EGABA_A_R-related due to the low HCO_3_^-^ conductance of GABA_A_R in embryonic spinal MNs. This is mentioned in the Introduction (see Bormann et al., 1987). A new reference is now included (Gao and Ziskind-Conhaim, 1995).

4) It is also not clear why the authors are indicating the prolonged relaxation of synaptic inhibitory events as a "compensation" (title, Abstract etc etc) since this is simply due to the higher intracellular concentration of Cl−, that strongly influence the duration of GABAergic and glycinergic synaptic inhibition while affecting the reverse potential.

We understand this ambiguity and we have tried to better introduce this point in the Introduction section.

How are the average mIPSC (recorded events) charges when considering the reduced amplitudes?

We have now added the charge measurements, which do not display any difference between SOD and WT MNs. This is indicated in the Results section.

5) The authors report an exciting finding that EGABA is 10 mV more depolarizing in SOD motor neurons. Since motor neurons express tonic GABAA conductance, this conductance can make cells more excitable and promote their spontaneous firing. A similar phenomenon has been previously reported in hippocampal interneurons. The authors may consider the depolarizing tonic GABAA conductance in SOD motor neurons.

After the reviewers’ comment, we have performed new experiments in which we have measured the tonic glycine and GABA conductance. We found a tonic conductance essentially for glycine and no difference between SOD and WT MNs. This finding is both included and discussed in the Results section.